# T-ALL leukemia stem cell 'stemness' is epigenetically controlled by the master regulator SPI1

Haichuan Zhu[1,2,3], Liuzhen Zhang[1,2,3], Yilin Wu[1,2,3], Bingjie Dong[1,2,3], Weilong Guo[1,2,3], Mei Wang[1,2,3], Lu Yang[1,2,3], Xiaoying Fan[1,2,3], Yuliang Tang[2,4], Ningshu Liu[5], Xiaoguang Lei[2,4], Hong Wu[1,2,3]*

[1]The MOE Key Laboratory of Cell Proliferation and Differentiation, School of Life Sciences, Peking University, Beijing, China; [2]Peking-Tsinghua Center for Life Sciences, Peking University, Beijing, China; [3]Beijing Advanced Innovation Center for Genomics, Peking University, Beijing, China; [4]Department of Chemical Biology, College of Chemistry and Molecular Engineering, Peking University, Beijing, China; [5]Drug Discovery Oncology, Bayer Pharmaceuticals, Berlin, Germany

**Abstract** Leukemia stem cells (LSCs) are regarded as the origins and key therapeutic targets of leukemia, but limited knowledge is available on the key determinants of LSC 'stemness'. Using single-cell RNA-seq analysis, we identify a master regulator, SPI1, the LSC-specific expression of which determines the molecular signature and activity of LSCs in the murine *Pten*-null T-ALL model. Although initiated by PTEN-controlled β-catenin activation, *Spi1* expression and LSC 'stemness' are maintained by a β-catenin-SPI1-HAVCR2 regulatory circuit independent of the leukemogenic driver mutation. Perturbing any component of this circuit either genetically or pharmacologically can prevent LSC formation or eliminate existing LSCs. LSCs lose their 'stemness' when *Spi1* expression is silenced by DNA methylation, but *Spi1* expression can be reactivated by 5-AZ treatment. Importantly, similar regulatory mechanisms may be also present in human T-ALL.
DOI: https://doi.org/10.7554/eLife.38314.001

*For correspondence:
Hongwu@pku.edu.cn

## Introduction

Acute T cell lymphoblastic leukemia (T-ALL) is an aggressive hematological malignancy caused by the accumulation of genetic mutations and altered signaling pathways that affect normal T cell development (*Belver and Ferrando, 2016*; *Ferrando and López-Otín, 2017*; *Girardi et al., 2017*). Current treatment for T-ALL includes high-intensity combination chemotherapies. However, such treatment may cause short- and long-term side effects, and up to 20% of pediatric and 40% of adult T-ALL patients relapse (*Atak et al., 2013*; *Girardi et al., 2017*). Leukemia stem cells (LSCs) are considered to be one of the main causes of drug resistance and therapeutic relapse (*Batlle and Clevers, 2017*; *Blackburn et al., 2014*; *Chiu et al., 2010*; *Visvader, 2011*). Like hematopoietic stem cells, LSCs can self-renew and differentiate into leukemic blast cells (*Bonnet and Dick, 1997*; *Reya et al., 2001*), which makes them ideal candidates for high-efficiency and low-toxicity targeted therapies. However, many questions related to the control mechanisms of LSCs and cancer stem cells (CSCs) in general remain unanswered.

One question is how CSCs maintain 'stemness'. Although many driver mutations and dysregulated pathways have been identified in cancers, these are unlikely to be the only mechanisms that maintain CSC 'stemness', since the same driver mutations or dysregulated pathways are also present in most cancer cells. One good example is the *Pten*-null T-ALL model that we have generated by the conditional deletion of the *Pten* tumor suppressor gene in fetal liver hematopoietic stem

cells (*Guo et al., 2008*). In this model, LSCs are enriched in the Lin⁻CD3⁺KIT$^{mid}$ cell subpopulation; these cells are self-renewable and responsible for T-ALL initiation and drug resistance (*Guo et al., 2008*; *Guo et al., 2011*; *Schubbert et al., 2014*). However, since both LSC-enriched and leukemic blast subpopulations share similar genetic alterations, including *Pten* loss and *Tcra/d-Myc* transloca-tion (*Guo et al., 2008*), these driver mutations are unlikely to determine LSC 'stemness'. Further-more, treating the *Pten*-null T-ALL model with PI3K inhibitors is effective only before the onset of leukemia, not after leukemia is already underway (*Guo et al., 2011*; *Schubbert et al., 2014*), sug-gesting that this driver mutation is not responsible for the maintenance of LSC 'stemness' once it has been generated.

A related question is how CSCs lose 'stemness' and whether this process is unidirectional or reversible. Such plasticity or reversibility may contribute to some of the conflicting results in the liter-ature regarding the nature and frequency of CSCs (*Batlle and Clevers, 2017*). As small-molecule inhibitors of epigenetic modifiers have been developed and applied to cancer treatments (*Topper et al., 2017*), understanding the nature of CSC maintenance may bear important clinical implications.

Using the *Pten*-null T-ALL model, we identify a master regulator, SPI1, and a β-catenin-SPI1-HAVCR2 regulatory circuit that are responsible for LSC 'stemness' maintenance. This 'stemness' maintenance circuit is initiated by the leukemogenic driver mutations, that is, PTEN loss and PI3K-mediated β-catenin activation, but after it is formed, it becomes independent of the driver mutation and the associated PI3K pathway. Furthermore, SPI1's LSC-specific expression is silenced by DNA methylation, resulting in the loss of LSC 'stemness'. Our study also provides the fate mapping of leu-kemia development from LSCs to leukemic blasts at single-cell resolution and identifies potential novel targets for LSC-mediated therapies.

## Results

### Redefine heterogeneous LSCs at single-cell resolution

We reported previously that the LSC-enriched Lin⁻CD3⁺KIT$^{mid}$ subpopulation in the *Pten*-null T-ALL model contains heterogeneous cells, of which 30% are MYC low, rapamycin- and JQ1 (a BRD4 inhibi-tor)-resistant, and relatively quiescent in terms of cell cycle (*Guo et al., 2008*; *Schubbert et al., 2014*) (*Figure 1—figure supplement 1A*). To further define this heterogeneous subpopulation, we isolated LSC-enriched and blast subpopulations for RNA-seq analysis (*Figure 1—figure supplement 1B*, upper panel) and identified one module with a LSC$^{high}$-Blast$^0$ expression pattern by Weighted Gene Co-expression Network Analysis (WGCNA) (*Zhang and Horvath, 2005*) (*Figure 1A*, yellow module). Approximately, 45% of the genes in this module encode membrane proteins such as *Havcr2* (HAVCR2) and *Itgax* (ITGAX) (*Figure 1B–C*). Although *Havcr2* and *Itgax* are only expressed in the LSC-enriched subpopulation, the expression levels of these genes vary among different isolates (*Figure 1C*), which may reflect the heterogeneity of the LSC-enriched subpopulation. The cell surface expression of HAVCR2 and ITGAX, as measured by FACS analysis, are highly correlated and can fur-ther separate the previously identified Lin⁻CD3⁺KIT$^{mid}$ LSC-enriched subpopulation into several sub-groups (*Figure 1D*, upper panel), among which the HAVCR2$^{high}$ or HAVCR2$^{high}$ ITGAX$^{high}$ subgroups are most abundant in the thymus, the critical organ for T cell development and T-ALL ini-tiation (*Guo et al., 2008*;*Guo et al., 2011*) (*Figure 1D*, lower panel).

To determine whether these heterogeneous groups are organized hierarchically from LSCs to blasts during T-ALL development, we conducted single-cell RNA-seq analysis and identified four subgroups (*Figure 1E*; *Figure 1—figure supplement 1B*, lower panel; *Figure 1—figure supple-ment 2*). Pseudotime analysis (*Trapnell et al., 2014*) further indicates that LSCs follow a continuous developmental path towards blasts, progressing from HAVCR2$^{high}$ through HAVCR2$^{mid}$ and HAVCR2$^{low}$ to blasts (*Figure 1F*), which can also be visualized by pseudotime analysis of *Havcr2* and *Itgax* expression (*Figure 1G*). Collectively, these results confirm the heterogeneity of the previously identified LSC-enriched subpopulation and provide fate mapping of LSC differentiation into blasts at single-cell resolution.

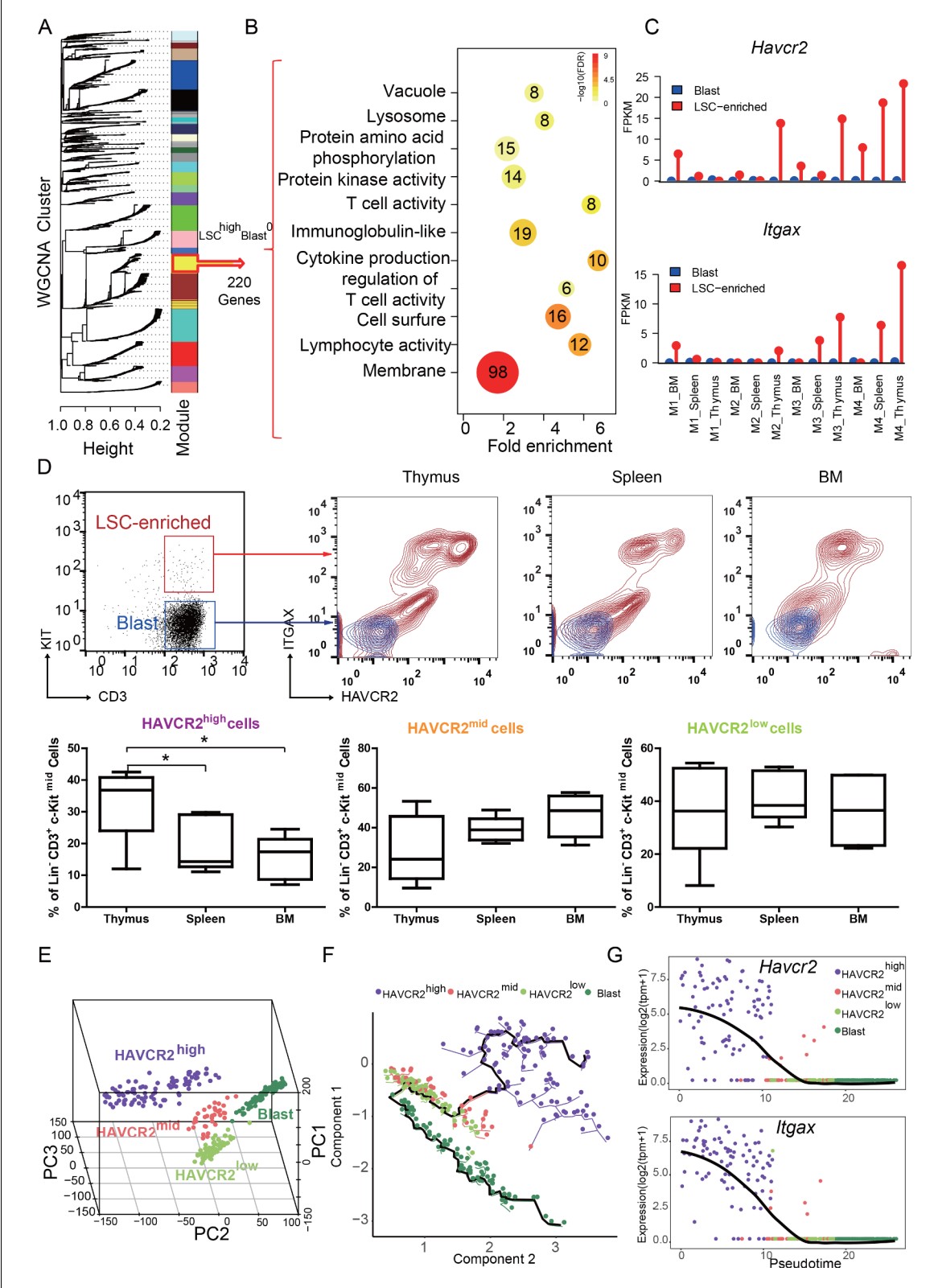

**Figure 1.** HAVCR2 redefines a heterogeneous LSC-enriched subpopulation at single-cell resolution (**A**) WGCNA analysis for the bulk RNA-seq of LSC-enriched and leukemic blast subpopulations. The yellow module contains 220 genes that are preferentially expressed in the LSC-enriched subpopulation (LSC^high-Blast^0); (**B**) Gene Ontology (GO) analysis of LSC-enriched genes in the yellow module; (**C**) *Havcr2* and *Itgax* are specifically expressed in LSC-enriched (red) but not in leukemic blast (blue) subpopulations isolated from the indicated hematopoietic organs of M1-M4 *Pten*-null

*Figure 1 continued on next page*

*Figure 1 continued*

T-ALL mice; (D) Upper panel: FACS plots are overlaid to show the differential expression of HAVCR2 and ITGAX in the LSC and blast subpopulations. The previously defined Lin⁻CD3⁺KIT$^{mid}$ LSC-enriched subpopulation (in the red box in the left panel) can be further separated into several subgroups based on the expression of the cell-surface markers HAVCR2 and ITGAX. The Lin⁻CD3⁺KIT⁻ leukemic blast subpopulation (in the blue box in the left panel) does not express HAVCR2 or ITGAX. Lower panel: Quantitative measurement of the HAVCR2$^{high}$, HAVCR2$^{mid}$ and HAVCR2$^{low}$ subgroups in different hematopoietic organs from *Pten*-null T-ALL mice (n = 5; *, p<0.05). The HAVCR2$^{high}$ subgroup is enriched in the thymus; (E) PCA analysis of the single-cell transcriptome shows four subgroups, labeled in different colors. Cells from two independent mice are indicated by different shapes; (F) Pseudotime analysis shows the expression profiles of T-ALL cells in 2-D component space. The solid black line shows the main differentiation path from HAVCR2$^{high}$ (purple) to blasts (dark green); (G) Pseudotemporal ordering of single cells based on *Havcr2* or *Itgax* expression. BM: bone marrow.

DOI: https://doi.org/10.7554/eLife.38314.002

The following figure supplements are available for figure 1:

**Figure supplement 1.** A schematic illustration of procedures used for Bulk and single cell RNAseq analysis.

DOI: https://doi.org/10.7554/eLife.38314.003

**Figure supplement 2.** Quality control of single cell RNAseq analysis.

DOI: https://doi.org/10.7554/eLife.38314.004

## The HAVCR2$^{high}$ subgroup contains the vast majority of LSC activity

Single-cell transcriptome analysis indicates that the HAVCR2$^{high}$ subgroup is enriched in the hematopoietic stem cell/late progenitor pathways and is relatively quiescent, while the blast subpopulation is enriched in *Myc* lymphoma pathways and active in the cell cycle (*Figure 2A*). Consistent with this observation, the HAVCR2$^{high}$ subgroup also has the lowest c-MYC level among the four subgroups (*Figure 2B*), suggesting that the HAVCR2$^{high}$ subgroup may contain the MYC$^{low}$ cells within the

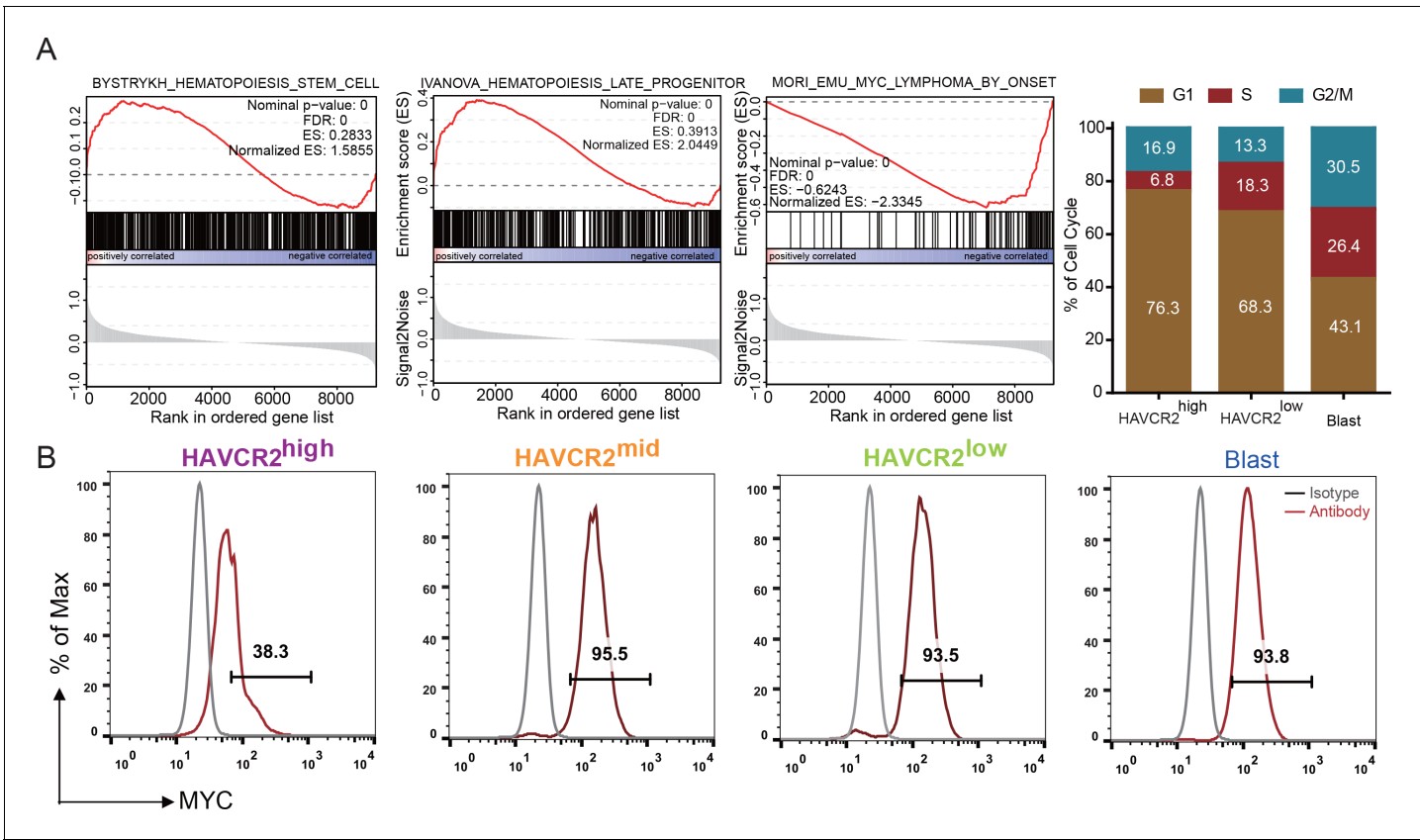

**Figure 2.** Cells in the HAVCR2$^{high}$ subgroup are in a quiescent cell cycle state. (A) Left panel: GSEA analysis shows signaling pathways enriched in the HAVCR2$^{high}$ and blast subpopulations. Right panel: Percentage of cells in each phase of the cell cycle based on single-cell RNA-seq; (B) Intracellular FACS analyses of MYC levels in the HAVCR2$^{high}$, HAVCR2$^{mid}$, HAVCR2$^{low}$ and blast subgroups. Gray line: isotype control.

DOI: https://doi.org/10.7554/eLife.38314.005

previously defined Lin⁻CD3⁺KIT^mid LSC-enriched subpopulation (**Guo et al., 2008**; **Schubbert et al., 2014**).

To determine whether HAVCR2^high cells are true LSCs, we performed limiting dilution and bone marrow transplantation analyses using 10 to 1000 bone marrow cells from the Lin⁻CD3⁺KIT^mid, Lin⁻CD3⁺KIT^mid HAVCR2^high, and Lin⁻CD3⁺KIT^mid HAVCR2^low subgroups (**Figure 3A**). Cells from the HAVCR2^high subgroup have the highest leukemia-initiating capacity—nearly every Lin⁻CD3⁺KIT^mid HAVCR2^high cell is capable of inducing T-ALL development, compared to 1/14 of the cells in the Lin⁻CD3⁺KIT^mid subgroup and 1/28 of the cells in the Lin⁻CD3⁺KIT^mid HAVCR2^low subgroup (**Figure 3B**). Consistent with these findings, cells from the HAVCR2^high subgroup can also induce

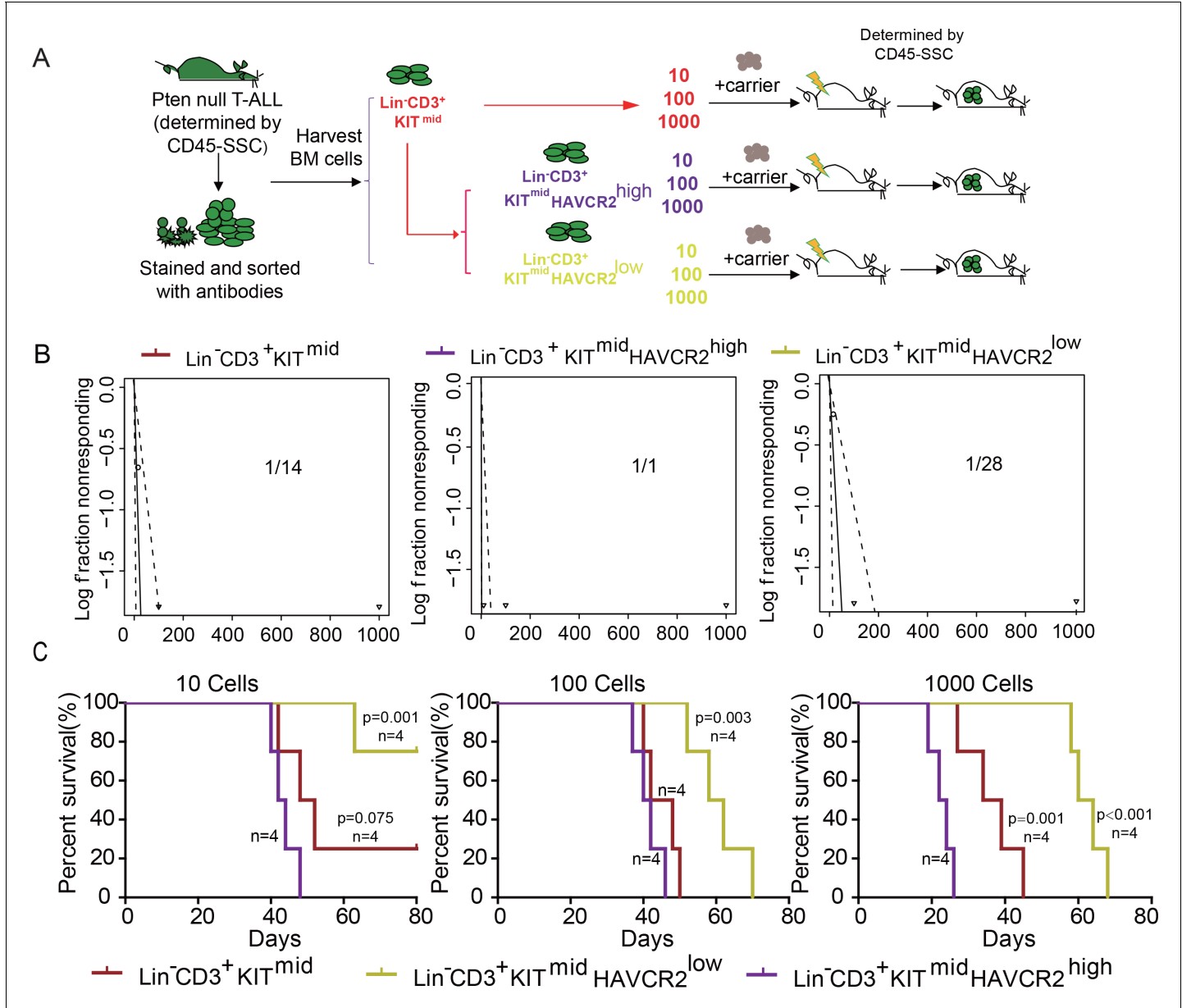

**Figure 3.** The HAVCR2^high subgroup contains the vast majority of LSC activity. (**A**) Schematic illustrating the cell isolation, limiting dilution and transplantation procedures used for testing LSC activity as described in Guo et al. (**Guo et al., 2008**); (**B**) LSC frequencies were calculated for each subgroup according to Hu et al. (**Hu and Smyth, 2009**);(**C**) Survival curves showing LSC activity in each of the sorted subgroups upon transplantation (n = 4). Student's *t*-test was used to calculate the p-value.
DOI: https://doi.org/10.7554/eLife.38314.007

T-ALL lethality much earlier than cells from the other two subgroups (*Figure 3C*). Thus, HAVCR2 is a novel surface marker for the isolation of pure LSCs, and the HAVCR2[high] subgroup represents the true LSC population in the *Pten*-null T-ALL model (*Table 1*).

## SPI1 is the master regulator of LSC signature genes

The identification of HAVCR2[high] cells as the true LSC population allows us to define the key determinant for LSC 'stemness'. We used network component analysis (*Tran et al., 2012*), in which the activity of transcription factors can be deduced based on the expression levels of their target genes. Among the predicted transcription factors (*Liberzon et al., 2011*) that may control the expression of HAVCR2[high] LSC signature genes, SPI1 scores the highest (data not shown). Importantly, approximately 70% of the HAVCR2[high] LSC signature genes overlap with SPI1 target genes identified during T cell development (*Zhang et al., 2012*)(*Figure 4A*). Therefore, we decided to focus our subsequent analysis on SPI1.

Since the HAVCR2[high] MYC[low] phenotype signifies LSCs, we first examined the correlation of *Spi1*, *Havcr2* and *Myc* expression in HAVCR2[high] and blast cells. The pseudotemporal ordering of the single-cell RNA-seq data and the FACS analyses demonstrate that *Spi1* expression is highest in the HAVCR2[high] subgroup, which is opposite to the differential expression of *Myc* (*Figure 4B–C*).

We further investigated whether SPI1 could transcriptionally regulate *Havcr2* and *Myc* expression by conducting SPI1 ChIP-qPCR analysis on *Spi1-Egfp* stably transformed blasts, using *Egfp*-transfected blasts as a control (*Figure 4—figure supplement 1*). SPI1 binds strongly to *Havcr2* promoter region 2 (*Zhu et al., 2015*) (*Figure 4D*) and the *Tcra* enhancer (EA) region in the translocated *Tcra/d-Myc* allele (*Figure 4E*), as well as to the E2 region of the WT allele (*Shi et al., 2013*)(*Figure 4F*), suggesting that it may have regulatory effects on both genes. The overexpression of *Spi1* in T-ALL blast cells significantly increases the expression of *Havcr2* and other known SPI1 target genes, such as *Itgax* and *Lmo2* (*Champhekar et al., 2015*; *Turkistany and DeKoter, 2011*; *Yashiro et al., 2017*), but downregulates *Myc* mRNA and protein levels (*Figure 4G–H*). In contrast, *SPI1* knockdown in a human T-ALL cell line downregulates the expression of SPI1 target genes but upregulates *MYC* expression (*Figure 4I*). Importantly, the positive correlation between *SPI1* and the expression of *HAVCR2* as well as that of SPI1 target genes such as *ITGAX* and *LMO2* can be found in human T-ALL datasets (*Liu et al., 2017*; *Van Vlierberghe et al., 2011*)(*Figure 5*), suggesting that the regulation of HAVCR2 expression by SPI1 could play an important role in human T-ALLs.

## SPI1 is essential for LSCs 'stemness' and T-ALL development

SPI1 is an ETS domain-containing transcription factor critical for early T cell progenitor function (*Zhang et al., 2012*), and its overexpression or translocation induces T progenitor cell proliferation and blocks differentiation (*Anderson et al., 2002*; *Seki et al., 2017*), similar to the effects we

**Table 1.** The biological properties of the newly defined HAVCR2[high] LSC subgroup in comparison to other subgroups in the *Pten*-null T-ALL model.

| Cell type | HAVCR2[High] | HAVCR2[mid] and HAVCR2[low] | Blasts |
|---|---|---|---|
| MYC | low | high | high |
| Rapamycin | resistance | sensitive | sensitive |
| JQ1 | resistance | sensitive | sensitive |
| BrdU | low | high | high |
| Surface marker | KIT[mid] | | KIT[-] |
| | HAVCR2[high]/ITGAX[high] | HAVCR2[mid/low]/ITGAX[mid/low] | HAVCR2[-]/ITGAX[-] |
| β-catenin activity | high | medium | low |
| LIC activity | 1/1 | 1/28 | 1/10^4–10^5 |
| Pathway | Stem/progenitor | | Myc/lymphoma |

DOI: https://doi.org/10.7554/eLife.38314.006

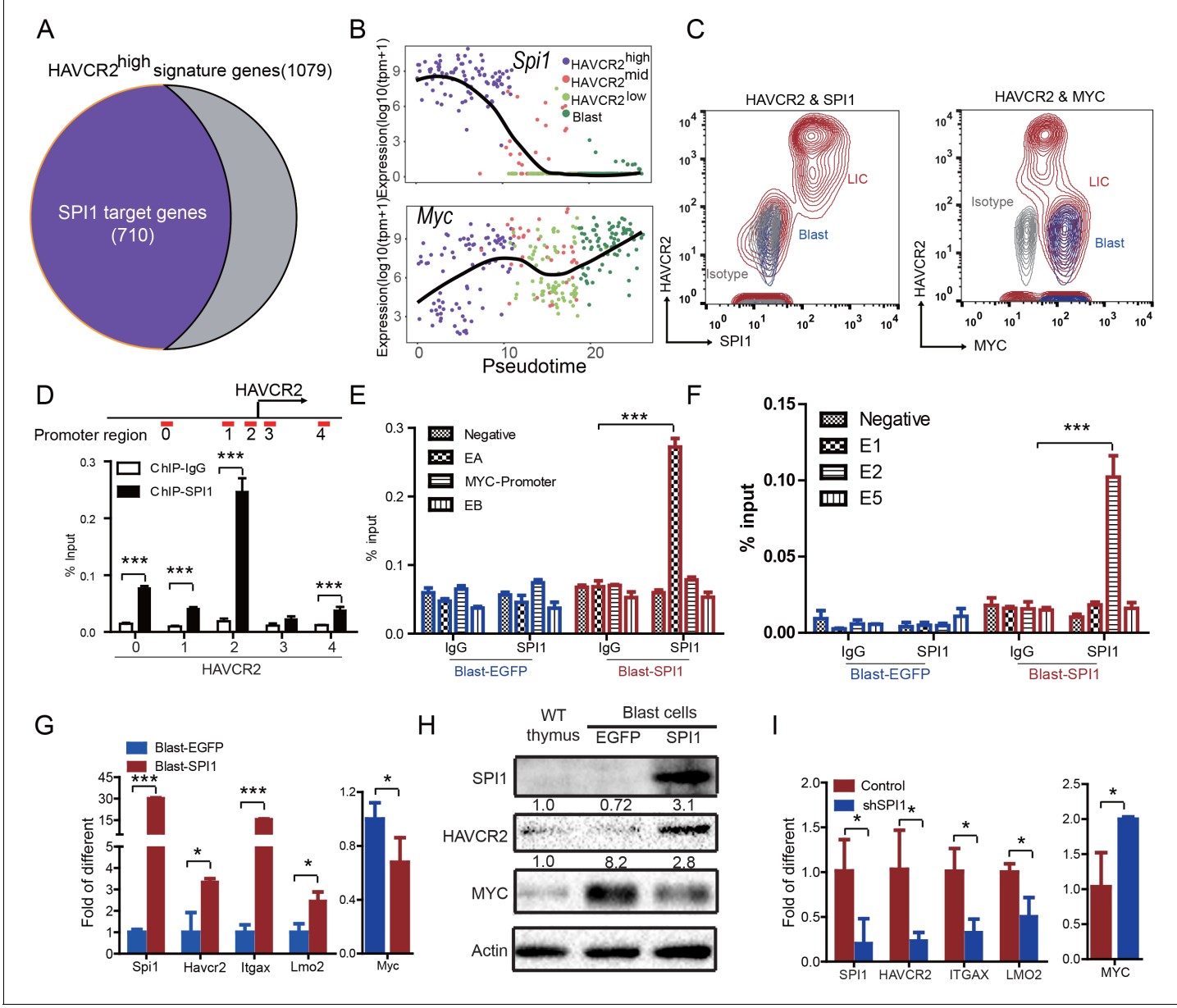

**Figure 4.** SPI1 is the master regulator of LSC signature genes and controls HAVCR2 and c-MYC expression. (**A**) Nearly 70% of the genes highly expressed in the HAVCR2high subgroup—the LSC signature genes—are potential SPI1 target genes (purple);(**B**) Pseudotemporal ordering of single cells based on *Spi1* or *Myc* expression; (**C**) FACS analysis shows the correlation of HAVCR2 cell surface expression and intracellular SPI1 and c-MYC levels in the LSC-enriched (Lin-CD3+KITmid; red) and blast (Lin-CD3+KIT-; blue) subpopulations. Gray, isotype control; (**D–E**) ChIP-qPCR analysis identifies SPI1 binding regions in the HAVCR2 promoter (left) and T*cra/d* enhancer A(EA) region (right), using Blast-SPI1 cells; (**F**) ChIP analysis identifies a SPI1 binding site in the endogenous *Myc* enhancer;(**G**) q-PCR shows the fold changes in *Havcr2*, *Itgax*, *Lmo2* and *Myc* expression between Blast-SPI1 cells (red) and Blast-EGFP cells (blue); (**H**) Western blotting shows the SPI1, HAVCR2 and c-Myc protein levels in WT thymus, Blast-EGFP and Blast-SPI1 cells. The fold changes relative to expression in the WT thymus are shown above each lane; (**I**) q-PCR analysis shows the fold changes in *HAVCR2*, *ITGAX*, *LMO2* and *MYC* expression in control shRNA (blue) and shSPI1 knockdown human T-ALL KE-37 cells (red); (**D–I**) All experiments were performed at least three independent times, and the data in D, E, F, G, and I are the means ± S.Ds; *p≤0.05; **p≤0.01; ***p≤0.001.

DOI: https://doi.org/10.7554/eLife.38314.008

The following figure supplement is available for figure 4:

**Figure supplement 1.** A schematic illustration of establishing lines expressing EGFP vector or EGFP-PU.1.

DOI: https://doi.org/10.7554/eLife.38314.009

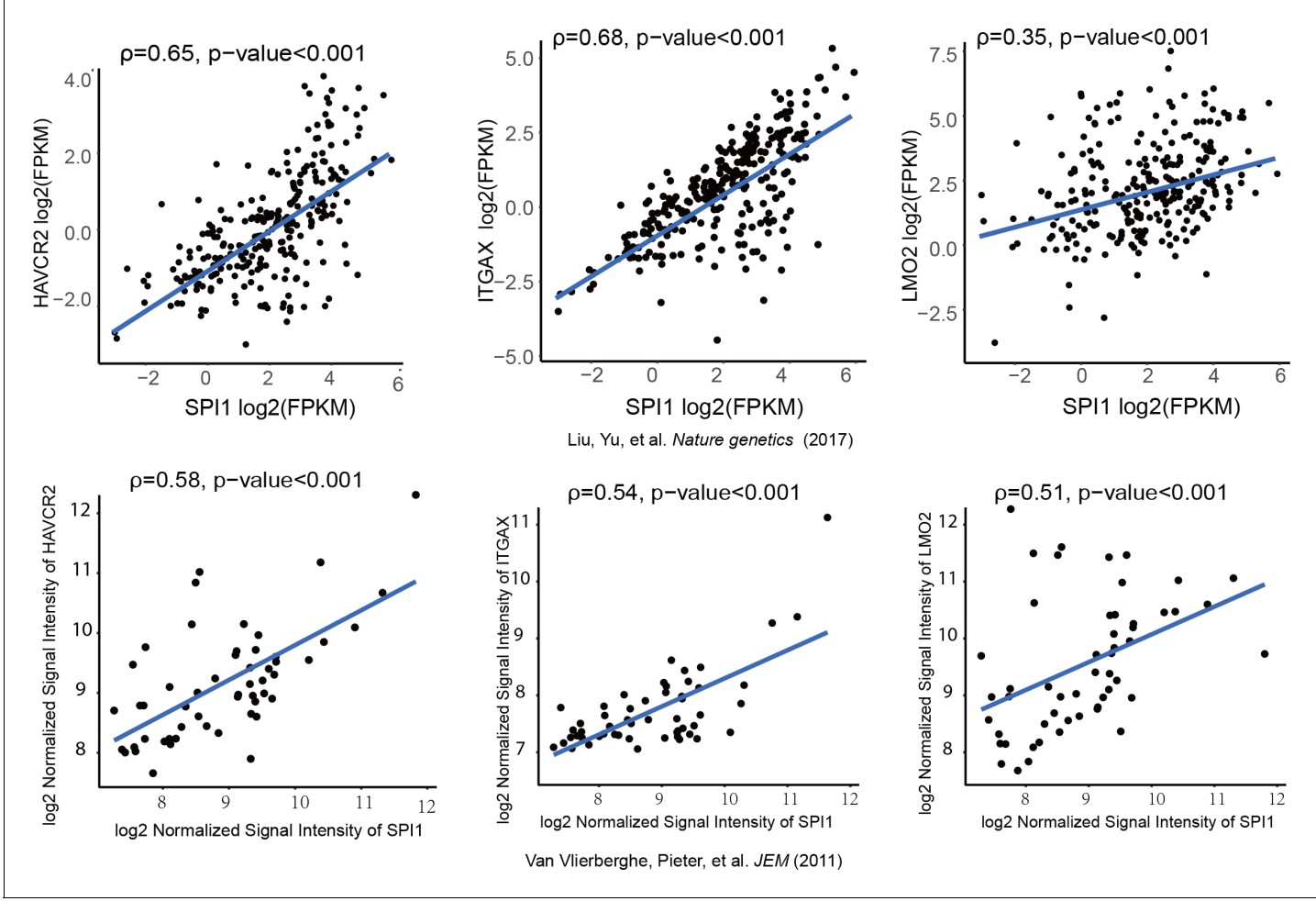

**Figure 5.** *SPI1* expression is positively correlated with *HAVCR2*, *ITGAX* and *LMO2* expression in human T-ALL. Correlation of SPI1 expression with HAVCR2, ITGAX and LMO2 expression in two different cohorts of human T-ALL samples, ρ: Spearman's rank correlation coefficient, p-value: p-value of Spearman's rank correlation test.

DOI: https://doi.org/10.7554/eLife.38314.010

observed in the *Pten*-null T-ALL model. To functionally determine the role of SPI1 in LSC 'stemness' and T-ALL development, we conditionally deleted *Spi1* in the *Pten*-null T-ALL model. Kaplan-Meier survival analysis shows that the lethality caused by T-ALL is delayed proportionally to the numbers of *Spi1* allele that are deleted (*Figure 6A*). The tissue architectures of the thymus and spleen appear normal, and no infiltrating leukemia cells can be detected in the liver of the mutant mice (dKO) (*Figure 6B*). FACS analyses also show the absence of HAVCR2[high] LSCs and CD3[+] blasts in the thymus, spleen and bone marrow (BM) of the dKO mice (*Figure 6C–D*). *Spi1* deletion can also restore spleen weight and organ morphology (*Figure 6B*; *Figure 6E*). Notably, the lethality seen in the compound homozygotes after 3 months is at least partially due to myeloid abnormalities, a known phenotype associated with SPI1 loss in the myeloid lineage (*Dakic et al., 2007*; *Rosenbauer et al., 2004*; *Steidl et al., 2006*)(data not shown).

To confirm that the absence of T-ALL in dKO mice is not due to a block in T cell development in the *Pten; Spi1*-null T progenitor cells (*Champhekar et al., 2015*; *Spain et al., 1999*), we first quantified CD3[+] T cells in the WT, *Pten*-null and dKO mice and found relatively normal numbers of CD3[+] cells in the dKO thymus (*Figure 6F*). We then crossed dKO mice with mice of the *Rosa26[loxp-stop-loxp]-LacZ* reporter line so that LacZ expression could be used to trace the behavior of cells with Cre-mediated deletion of *Pten* and *Spi1* (*Guo et al., 2008*; *Guo et al., 2011*). Our FACS-Gal analysis shows that like LacZ[-] WT cells (blue), LacZ[+] dKO cells (red) in the same animals can undergo

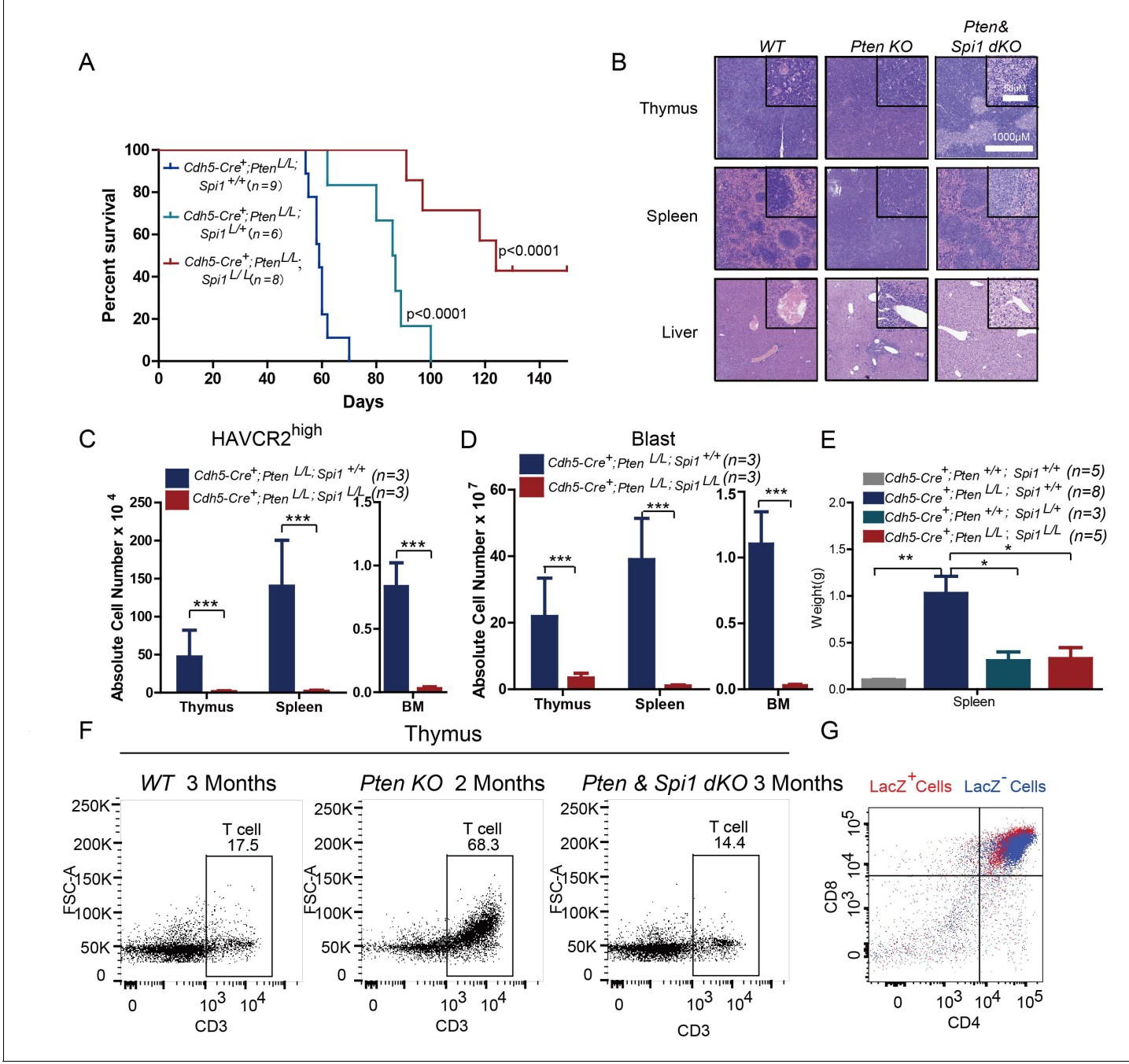

**Figure 6.** SPI1 is essential for LSC formation and T-ALL development. (**A**) Survival curves for *Cdh5-Cre+;Pten^L/L* T-ALL model mice (blue line) with heterozygous (*Cdh5-Cre+;Pten^L/L;Spi1^L/+*; green line) or homozygous (*Cdh5-Cre+;Pten^L/L;Spi1^L/L*; red line) *Spi1* conditional deletion; (**B**) HE-stained images of thymus, spleen and liver tissue from 2-month-old mice with the indicated genotypes;(**C–D**) Comparison of the absolute number of HAVCR2^high and blast cells in each organ in 2-month-old *Cdh5-Cre+;Pten^L/L* (blue bars) and *Cdh5-Cre+;Pten^L/L;Spi11^L/L* (red bars) mice;(**E**) Comparison of spleen weights in the mice in B-C; (**F**) Representative FACS plots show CD3-positive T cells in the thymus of *WT*, *Pten*-null T-ALL and *Pten/Spi1* double knockout mice. *WT* and *Pten/Spi1* double knockout mice were 3 months old, and *Pten*-null T-ALL mice were 2 months old. n = 3; (**G**) FACS-Gal analysis of T cell development in the thymus of *Pten/Spi1* double knockout mice. LacZ+ cells (red dots) and LacZ− cells (blue dots) from the same sample are overlaid. C-D, the data are presented as the means ± S.Ds; *p≤0.05; **p≤0.01; ***p≤0. 001.The bars in the HE images and inserts represent 1000 μM and 50 μM, respectively.

DOI: https://doi.org/10.7554/eLife.38314.011

differentiation to become CD4⁺CD8⁺ double-positive T cells (*Figure 6G*). These results suggest that PI3K activation can rescue the T cell developmental block in *Spi1*-null T cell progenitors (*Champhekar et al., 2015*; *Spain et al., 1999*), similar to the findings in our previous report on *Pten; Rag*-null mice (*Guo et al., 2011*).Therefore, SPI1 is essential for *Pten*-null LSC 'stemness' and T-ALL development.

## *Spi1* is upregulated at the ETP/DN1 stage during T cell development

The essential role of SPI1 in regulating LSC signature genes and 'stemness' prompted us to investigate how *Spi1* is regulated in the *Pten*-null T-ALL model. During T cell development, *Spi1*, with other T progenitor cell factors and growth factor receptors such as *Bcl11a*, *Lmo2*, *Flt3* and *Kit*, is highly expressed at the early T progenitor (ETP) and double-negative 1 (DN1) stage and is then immediately downregulated during T cell commitment (*Zhang et al., 2012*) (*Figure 7A*, upper panel). Interestingly, our pseudotemporal ordering of the single-cell RNA-seq data indicates that the expression patterns of *Spi1* and these factors and receptors are largely unchanged in the *Pten*-null T-ALL model compared to normal T cell development (*Figure 7A*, lower panels). Furthermore, these factors and receptors are highly expressed in the HAVCR2^{high} subgroup and downregulated in the HAVCR2^{mid} and HAVCR2^{low} subgroups, suggesting that HAVCR2^{high}SPI1^{high} LSCs may be generated at the ETP/DN1 stage (*Figure 7A*, low panels). Indeed, when we crossed *Spi1-GFP* reporter mice (*Nutt et al., 2005*) to *Pten*-null T-ALL model mice, we found that *Spi1-GFP* expression is significantly upregulated at the ETP/DN1 stage (*Figure 7B*).

## A β-catenin-SPI1-HAVCR2 regulatory circuit is required for *Spi1* upregulation and LSC 'stemness'

*β*-Catenin is an important transcription factor regulating *Spi1* expression in the T cell lineage (*Rosenbauer et al., 2006*). Previous works by us and others suggest that *β*−catenin is critical for

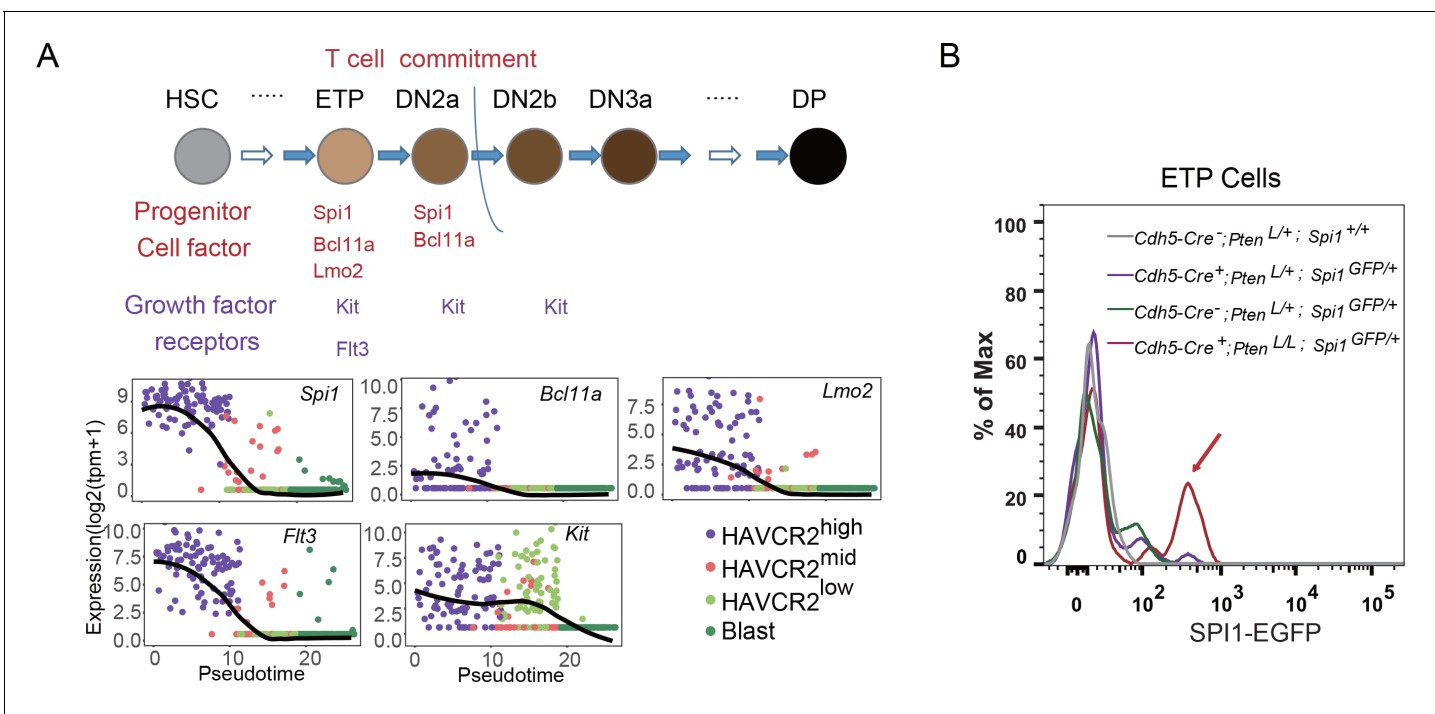

**Figure 7.** *Spi1* is upregulated at the ETP/DN1 stage during T cell development. (**A**) Upper panel: Diagram of progenitor cell factors and growth factor receptors involved in early T cell development, modified from (*Rothenberg et al., 2016*); lower panels: pseudotemporal ordering of single cells based on *Spi1, Bcl11a, Lmo2, Flt3* and *Kit* expression; (**B**) *Spi1-GFP* expression is upregulated in ETP/DN1progenitor cells from *Cdh5-Cre⁺;Pten^{L/L}; Spi1^{GFP/+}Pten* null (red line), compared to that in *Cdh5-Cre⁻;Pten^{+/L};Spi1^{+/+}* WT (gray line), *Cdh5-Cre⁺;Pten^{+/L};Spi1^{GFP/+}Pten* heterozygous (purple line) and *Cdh5-Cre⁻;Pten^{+/L};Spi1^{GFP/+}* WT GFP⁺ (green line) mice.

DOI: https://doi.org/10.7554/eLife.38314.012

LSC self-renewal (*Guo et al., 2008*) and RAG-dependent aberrant TCR rearrangement (*Dose et al., 2014*; *Guo et al., 2011*), a mechanism underlying the reoccurring *Tcra/d -Myc* translocation caused by PTEN loss or $\beta-$catenin activation observed in T-ALLs (*Guo et al., 2008 Kaveri et al., 2013*). Indeed, the overexpression of $\beta$-catenin in a human T-ALL cell line leads to the significantly increased expression of *SPI1* from its endogenous promoter and subsequently promotes the expression of its target gene *HAVCR2* but downregulates *MYC* expression (*Figure 8A*).

PTEN loss or PI3K/AKT activation is known to activate $\beta-$catenin by phosphorylating GSK-3$\beta$ and preventing GSK-3$\beta$-mediated $\beta-$catenin degradation (*Dan et al., 2008*; *Kikushige et al., 2015*; *Persad et al., 2001*). Although the HAVCR2^high, HAVCR2^low and blast subgroups have similar levels of P-GSK-3$\beta$ due to PTEN loss (*Figure 8B*, upper and lower panels), the HAVCR2^high subgroup has a much higher level of non-phospho-$\beta-$catenin (the active form of $\beta-$catenin) and SPI1 than HAVCR2-^low and blast subgroups *in vivo* (*Figure 8C and E*, upper and lower panels), suggesting that SPI1-mediated LSC formation may depend on mechanisms other than the oncogenic driver mutation PTEN loss.

HAVCR2 signaling can activate NF$\kappa$B and $\beta-$catenin and promote AML LSC formation and self-renewal (*Kikushige et al., 2015*). Since we identified HAVCR2 as the SPI1 target gene, we hypothesized that HAVCR2 signaling may in turn activate *Spi1* expression and promote T-ALL LSC formation. Intracellular FACS analyses show that among the four subgroups, the HAVCR2^high subgroup, which has the highest *Spi1* expression, also has the highest level of both phospho-p65 and non-phospho-$\beta-$catenin (*Figure 8D–E*, upper and lower panels), indicating that HAVCR2 signaling must play an important role in the hyperactivation of NF$\kappa$B and $\beta-$catenin. Consistent with this hypothesis, the genetic deletion of *Spi1* can prevent both HAVCR2^high LSC formation at the ETP/DN1 stage and T-ALL development (*Figure 8F*). The pharmacological inhibition of $\beta-$catenin activation by the novel tankyrase inhibitor BAY6060, but not the inhibition of PI3K activity by BAY1082439 alone (*Hill et al., 2017*), can also significantly reduce the number of HAVCR2^high LSCs *in vivo* in late-stage T-ALL (*Figure 8G*). Together, these results suggest that although *Spi1* upregulation is initiated by PTEN loss, SPI1-mediated LSC formation and 'stemness' are maintained by the $\beta-$catenin–SPI1-HAVCR2 regulatory circuit.

## LSCs loses their 'stemness' when *Spi1* expression is silenced by DNA methylation

How cancer stem cells lose 'stemness' and whether this process is unidirectional or reversible are currently unknown. Since *Spi1* expression is drastically reduced from the HAVCR2^high stage to the HAVCR2^low stage (*Figure 7A*, lower panel), we hypothesized that a *Spi1* silencing mechanism may explain the loss of LSC 'stemness' during differentiation. DNA methylation is one of the major epigenetic mechanisms in regulating gene expression during normal development. Although the global methylation patterns across the LSC signature and blast signature genes are similar (*Figure 9A–B*), the *Spi1* promoter is significantly hypomethylated in LSCs compared to blasts and normal T cell controls (*Figure 9C*). Consistently, the 4 CpG islands on the *Spi1* promoter (*Fernández-Nestosa et al., 2013*) are not methylated in the HAVCR2^high subgroup but gradually become methylated in the HAVCR2^mid and HAVCR2^low subgroups and are completely methylated in blasts (*Figure 9D*), which may explain the trend in *Spi1* expression and *Spi1*-controlled *Havcr2* and *Itgax* expression (*Figure 4B*, upper panel; *Figure 1G*). Conversely, treating leukemic blasts with the DNMT inhibitor 5-AZ can increase the expression of *Spi1* and its regulated LSC signature genes *in vitro* (*Figure 9E*) and induces the SPI1^+ and MYC^low subgroups *in vivo* (*Figure 9F*), demonstrating that *Spi1* expression is reversibly regulated by DNA methylation, which in turn regulates LSC signature gene expression.

To test the relevance of our findings to human T-ALL, we used two human T-ALL cell lines, KE-37 and CEM (*Burger et al., 1999*; *Tatetsu et al., 2007*). KE-37 expresses *SPI1* and *HAVCR2*, while CEM does not (*Figure 10A*), consistent with the methylation status of the *SPI1* promoter (*Figure 10B*). 5-AZ treatment can upregulate the expression of *SPI1* and its target *HAVCR2* but downregulate *c-MYC* expression in CEM cells, similar to the effects of our blast treatment, while no change can be detected in KE-37 cells (*Figure 10C*), demonstrating that *SPI1* expression is also regulated by DNA methylation in human T-ALL. To test whether the leukemogenic activity could be modulated by *SPI1* expression in human T-ALL cell lines, we injected placebo- or 5-AZ-treated CEM cells and monitored the T-ALL development induced by these cells *in vivo*. 5-AZ treatment

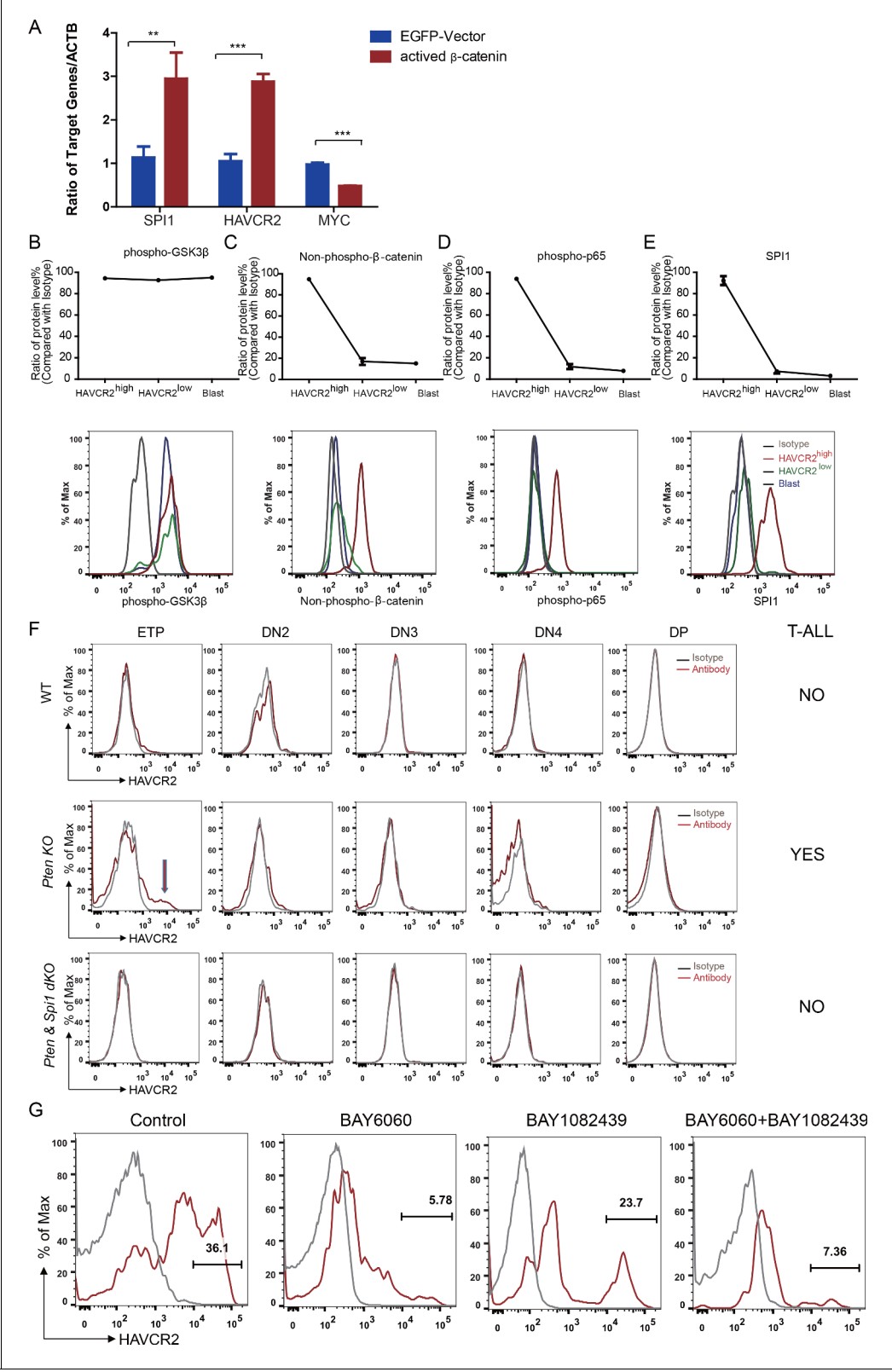

**Figure 8.** *Spi1* expression is maintained by *β*-catenin-SPI1-HAVCR2 regulatory circuit.  (**A**) q-PCR analysis of SPI1 and SPI1-regulated HAVCR2 and c-MYC expressions after the overexpression of active β-catenin in the Jurkat T-ALL cell line (red bars). The data are normalized to that of empty plasmid controls (blue bars); (**B–E**) Upper panels: quantitative intracellular FACS analyses of P-GSK-3β, non-phospho-β-catenin, P-p65 and SPI1 levels in the HAVCR2^high, HAVCR2^low and blast subgroups; lower panels: representative intracellular FACS analysis of P-GSK-3β, non-phospho-β-catenin, P-p65 and

*Figure 8 continued on next page*

*Figure 8 continued*

SPI1 levels in the HAVCR2$^{high}$, HAVCR2$^{low}$ and blast subgroups. Gray line, isotype control;(F) FACS analysis shows cells in the HAVCR2$^{high}$ subgroup at the ETP/DN1 stage, which are absent in WT and *dKO* mice; (G) Representative FACS plots show the number of cells in the HAVCR2$^{high}$ subgroup in the different drug treatment groups. The data in A, B, C, D and E are the means ± S.Ds of 3 independent tests; *p≤0.05; **p≤0.01; ***p≤0.001.
DOI: https://doi.org/10.7554/eLife.38314.013

significantly accelerated T-ALL development (*Figure 10D*). However, cell lines are not the best model system for studying LSC activity, and the essential role of SPI1 in regulating LSC activity in human T-ALL needs to be determined using patient samples and PDX models.

## Cotargeting oncogenic driver mutations and LSC 'stemness' maintenance circuit

We previously reported that treating *Pten*-null T-ALL model mice with a PI3K inhibitor is effective only at the preleukemia stage, not after leukemia has developed (*Guo et al., 2011*; *Blackburn et al., 2014*), suggesting the importance of cotargeting the LSC 'stemness' maintenance pathway once LSCs have been generated. Since SPI1 is essential for LSC formation and *SPI1* expression is regulated and maintained by the β-catenin-SPI1-HAVCR2 regulatory circuit, we hypothesized that cotargeting any component of this circuit with an anti-PI3K inhibitor may effectively eliminate existing T-ALL cells.

To test this hypothesis, we first treated age-matched leukemia-stage *Pten*-null T-ALL mice with DB1976 (*Figure 11—figure supplement 1A–B*), a compound known to specifically disrupt the interactions between SPI1 and its targets (*Antony-Debré et al., 2017*; *Munde et al., 2014*; *Stephens et al., 2016*). DB1976 can significantly inhibit the expression of *Havcr2* and other SPI1 target genes *in vitro* (*Figure 11A*) and reduce the number of HAVCR2$^{high}$ LSCs *in vivo* (*Figure 11B*, left panel), confirming that SPI1 is not only important for LSC formation but also for LSC maintenance. However, only when combined with a debulking anti-PI3K agent such as rapamycin (*Guo et al., 2008*) could DB1976 significantly reduce the leukemia burden, as demonstrated by the nearly complete absence of leukemic blasts in the hematopoietic organs (*Figure 11B*, right panel). Consequently, combination treatment can markedly prolong the animal lifespan (*Figure 11C*), restore the spleen weight and morphology, and eliminate infiltrating leukemic cells in the lung, kidney and liver without a significant change in animal body weight (*Figure 11D–E*; *Figure 11—figure supplement 1C*). Similar results were obtained when we replaced DB1976 and rapamycin with BAY6060 and BAY1082439, respectively (*Figure 11D–E*; *Figure 11—figure supplement 1D*). BAY1082439 can inhibit PI3Kδ, which is essential for *Pten-null* leukemia (*Subramaniam et al., 2012*), at nanomolar concentrations (*Antony-Debré et al., 2017*). The inhibition of tankyrase by BAY6060 can significantly reduce β-catenin activity and consequently decrease *Spi1* expression and the number of HAVCR2$^{high}$ LSCs *in vivo* (*Figure 12A*). In combination, BAY6060 and BAY1082439 could significantly prolong the animal lifespan and almost completely eliminate LSCs and blasts (*Figure 12B*, *Figure 11E* and *Figure 8G*).

Compared with β-catenin and SPI1, HAVCR2 may be a better therapeutic target as it is normally not expressed in hematopoietic stem and progenitor cells (*Kikushige et al., 2010*), and inhibition of HAVCR2 would therefore be less toxic. An anti-HAVCR2 antibody has been used clinically in immunotherapy and in targeting AML LSCs (*Kikushige et al., 2010*; *Koyama et al., 2016*). When combined with rapamycin, the anti-HAVCR2 antibody showed a therapeutic effect similar to that seen for DB1976/rapamycin and BAY6060/BAY1082439 combinations (*Figure 11D–E* and *Figure 12C*; *Figure 11—figure supplement 1E*). Together, these results suggest that inhibiting any component in the β-catenin-SPI1-HAVCR2 regulatory circuit will inhibit LSC 'stemness' maintenance and lead to the effective elimination of HAVCR2-positive T-ALL cells in the presence of an effective debulking agent targeting the PI3K pathway, such as rapamycin or BAY1082439.

## Discussion

Our study suggests that two layers of control mechanisms may play essential roles in leukemogenesis (*Figure 13*). The first layer is driven by the loss of the PTEN tumor suppressor or the activation of the PI3K pathway, which leads to β-catenin activation, *Tcra/d-Myc* translocation and T-ALL

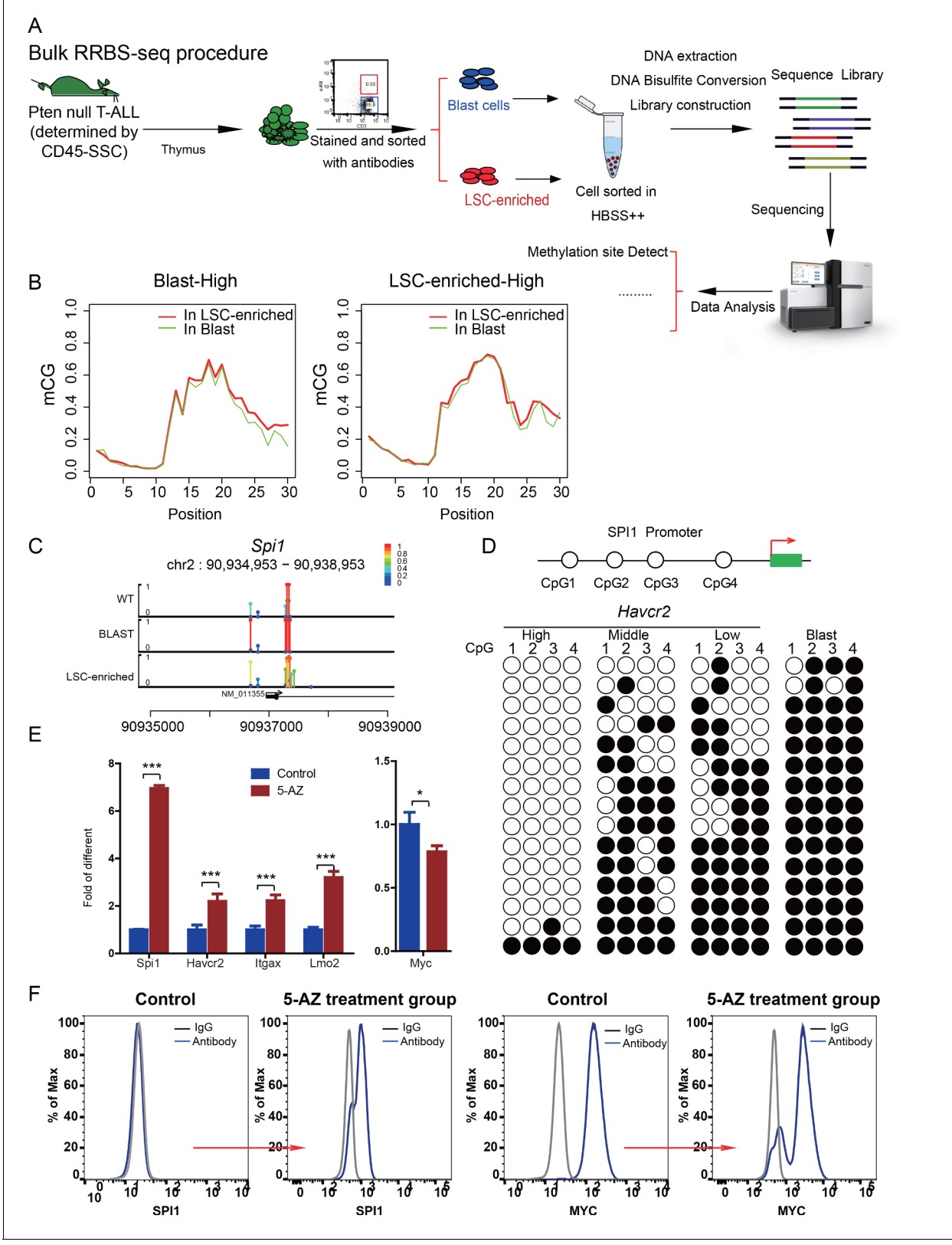

**Figure 9.** *Spi1* expression is controlled by DNA methylation **(A)** Schematic illustrating the procedures involved in cell isolation and RRBS analysis; **(B)** DNA methylation status of genes specifically expressed in the leukemic blast (left) and LSC-enriched (right) subpopulations; **(C)** *Spi1* promoter methylation status in normal T cells, LSC-enriched cells and blast-enriched cells.

DOI: https://doi.org/10.7554/eLife.38314.014

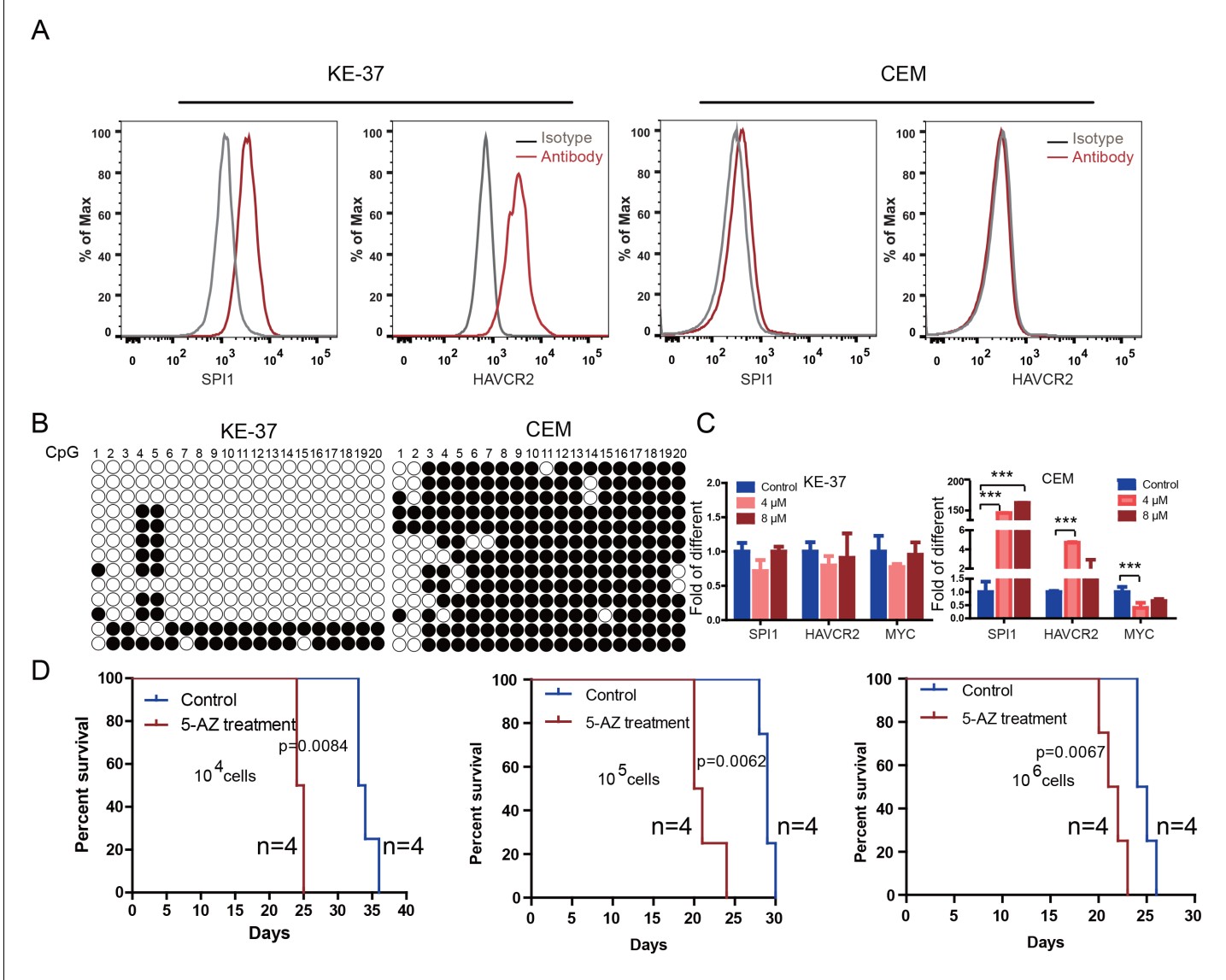

**Figure 10.** Human *SPI1* expression is silenced by DNA methylation (**A**) FACS analysis shows the surface expression of HAVCR2 and the intracellular level of SPI1 in the human T-ALL cell lines KE-37 and CEM.(**B**) Methylation status of CpG islands in the *SPI1* promoter in the human T-ALL cell lines KE-37 and CEM; (**C**) q-PCR analysis of *SPI1*, *HAVCR2* and *Myc* expression in KE-37 and CEM cells without (blue) and with(pink and red) 5-AZ treatment *in vitro*; (**D**) Survival curves show T-ALL development by CEM cells without (blue) and with (red) 5-AZ treatment upon transplantation (n = 4; *t*-test). The data in Care the means ± S.Ds of 3 independent tests; *p≤0.05; **p≤0.01; ***p≤0.001.

DOI: https://doi.org/10.7554/eLife.38314.015

development. The second layer is controlled by the master regulator SPI1, which determines LSC signature gene expression and maintains LSC 'stemness' (*Figures 4–6*). SPI1 upregulation is initiated by PI3K-controlled β-catenin activation,while the LSC-specific expression of SPI1 is reinforced by the β-catenin-SPI1-HAVCR2 regulatory circuit (*Figure 8*). Once formed, LSCs are very sensitive to any perturbation of this regulatory circuit but are less dependent on the PI3K pathway, as inhibiting the PI3K pathway at the leukemia stage has little effect on the LSC number (*Figures 11–12*)(*Guo et al., 2008*; *Blackburn et al., 2014*). SPI1 is silenced by DNA methylation, which leads to the downregu-lated expression of LSC signature genes, the loss of LSC 'stemness' and leukemic differentiation (*Figures 9–10*). Although the PTEN loss and *Tcra/d-Myc* translocation in the first layer of the leuke-mogenesis mechanism are hardwired and present in both LSCs and leukemia blasts, the SPI1 expres-sion and maintenance in the second layer of the LSC 'stemness' mechanism is reversible and present

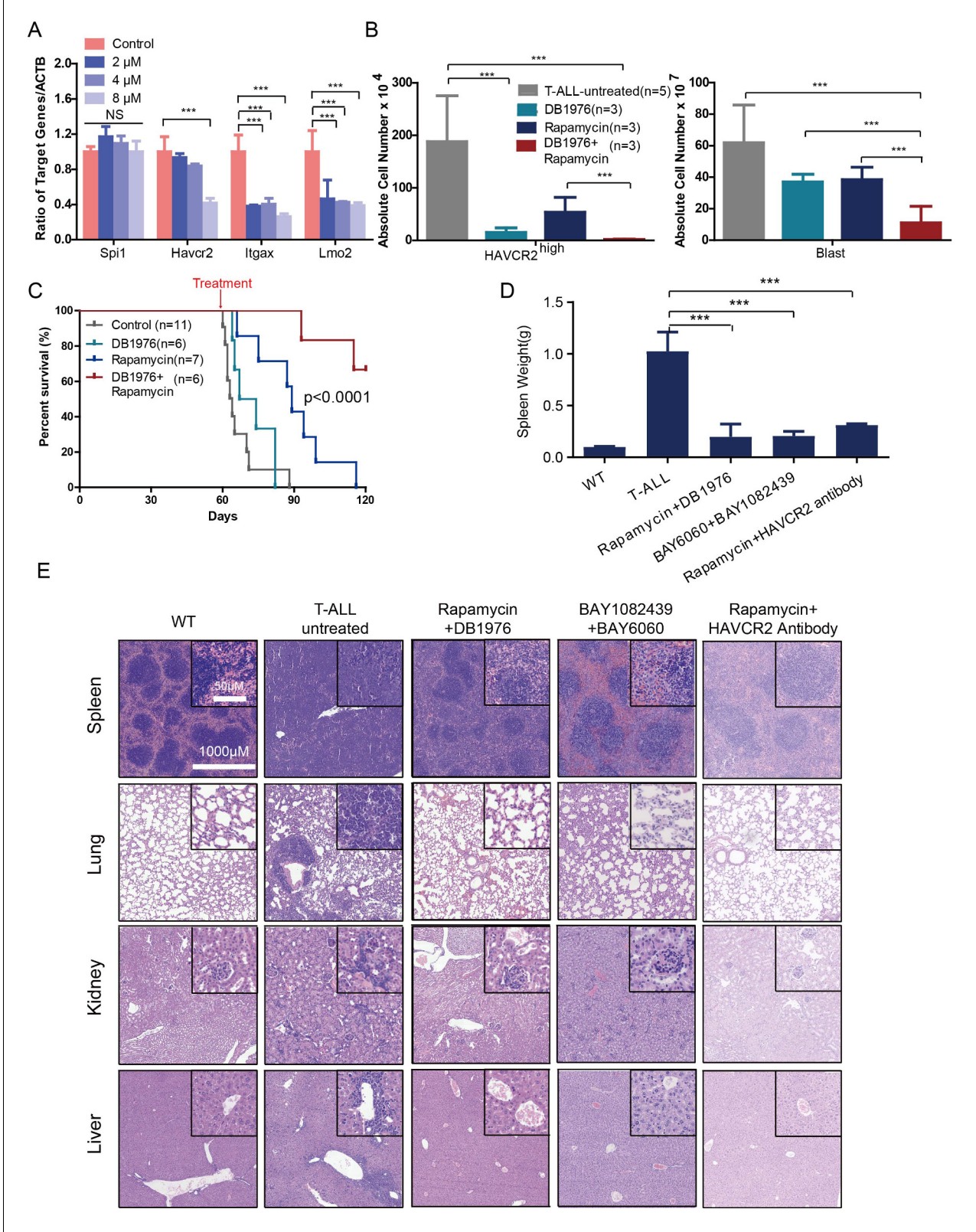

**Figure 11.** Cotargeting oncogenic driver mutations and the LSC 'stemness' maintenance circuit eliminated LSC and T-ALL cells  (A) q-PCR analysis of *Spi1* and *Spi1*-regulated *Havcr2*, *Itgax* and *Lmo2* expression after 24 hr of DB1976 treatment (blue bars). The data are normalized to that of the untreated controls (red bars); (B) A comparison of the absolute number of HAVCR2high and blast cells in the untreated (gray bars) and differently treated groups; (C) Survival curve of *Cdh5-Cre+;PtenL/L* model mice treated with DB1976 and rapamycin alone and in combination; (D) A comparison of the

*Figure 11 continued on next page*

*Figure 11 continued*

spleen weights of 2-month-old WT mice, untreated *Cdh5-Cre⁺;Pten^{L/L}* mice, and combination-treated mice upon euthanasia; (E) HE-stained images of spleen, lung, kidney and liver tissue from 2-month-old WT, untreated and combination-treated mice. A, B and D: the data are presented as the means ± S.Ds; ***p≤0.001; the bars in the HE images and inserts represent 1000 μM and 50 μM, respectively.

DOI: https://doi.org/10.7554/eLife.38314.016

The following figure supplement is available for figure 11:

**Figure supplement 1.** A schematic illustration of dosing schedules and treatment cohorts.

DOI: https://doi.org/10.7554/eLife.38314.017

only in LSCs (*Figure 13*). Similar two-layer control mechanisms may also be present in other types of cancer in which CSCs are known to play essential roles.

This two-layer model may have important implications for LSC-targeted therapies. First, targeting driver mutations or dysregulated pathways in the first layer may be sufficient for debulking the leukemia mass but not for eliminating LSCs unless the mechanism for maintaining LSC 'stemness' is simultaneously inhibited (*Figures 11–12*). Second, since the expression of the LSC master regulator SPI1 can be reversibly regulated by epigenetic mechanisms (*Figures 9–10*), this model would predict poorer outcomes if leukemia controlled by such a mechanism was treated with 5-AZ or similar agents and would suggest that the reactivation of SPI1 expression could be a potential mechanism for LSC-mediated therapeutic resistance.

A broad spectrum of epigenetic and genetic alterations has been found in virtually all cancer types. In certain cases, mutations within the epigenetic control machinery can influence global gene expression and cause subsequent cancer heterogeneity and clonal diversity; in other cases, epigenetic mechanisms may act on a specific transcription factor. Although we did not detect significant global methylation differences between LSC signature genes and blast signature genes, SPI1, the master regulator found in this study, is specifically methylated during differentiation from the HAVCR2^{high} to the HAVCR2^{low} phenotype (*Figure 9*), resulting in down regulating the expression of LSC signature genes. The mechanism that controls the specific methylation of SPI1 is currently unknown, but we predict that a similar mechanism may also regulate SPI1 silencing during T cell commitment (*Zhang et al., 2012*). The alteration of this silencing mechanism may lead to a block of T cell development and contribute to early progenitor type of T-ALL, such as ETP-T-ALL.

The identification of specific markers expressed only in LSCs is essential for isolating pure LSCs and studying their control mechanisms. Using an advanced single-cell sequencing technique, we identified HAVCR2 as an LSC-specific biomarker that can be used to isolate 'pure' LSCs, as determined by our limiting dilution and transplantation experiments (*Figures 1–3*). Most of the cell surface markers currently used to isolate LSCs or CSCs are irrelevant to the function of LSCs or CSCs. HAVCR2 is not just another biomarker but is an important regulator of the function of LSCs in *Pten*-null T-ALL. HAVCR2 is directly regulated by SPI1 and serves as an important component of the β-catenin-SPI1-HAVCR2 regulatory circuit, which is essential for maintaining the LSC-specific expression of SPI1 and LSC 'stemness' (*Figures 4–5* and *7–8*). HAVCR2 can also serve as an LSC-specific target (*Figures 11–12*); this finding is similar to that in a recent AML publication (*Kikushige and Akashi, 2012*; *Kikushige et al., 2015*).

PTEN and the PI3K/AKT/mTOR pathway controlled by PTEN are critical for the etiology of human T-ALL (*Gutierrez et al., 2009*; *Larson Gedman et al., 2009*; *Liu et al., 2017*; *Maser et al., 2007*; *Palomero et al., 2007*), and our study may illuminate the understanding and treatment of T-ALLs associated with PTEN loss or PI3K activation. We demonstrate that *SPI1* expression is upregulated by β-catenin and silenced by DNA methylation in human T-ALL cell lines, similar to the findings in the *Pten*-null T-ALL model (*Figures 9–10*). SPI1 also controls the expression of *HAVCR2* and other LSC signature genes in human T-ALL cell lines and clinical samples (*Figure 5*). However, whether the β-catenin-SPI1-HAVCR2 regulatory circuit also presents in human T-ALLs, especially the ETP T-ALL subtype, and determines LSC activity needs follow-up study using human T-ALL samples and PDX models. Such information may be used for the molecular classification of human T-ALLs, identifying human T-ALL LSCs and designing targeted treatment, as we showed in the mouse model. As HAVCR2^{high} LSCs can be detected in the peripheral blood of leukemic mice (our unpublished data), further investigation is worthwhile to explore the potential use of this approach as a noninvasive strategy for stratifying T-ALL and monitoring the treatment response.

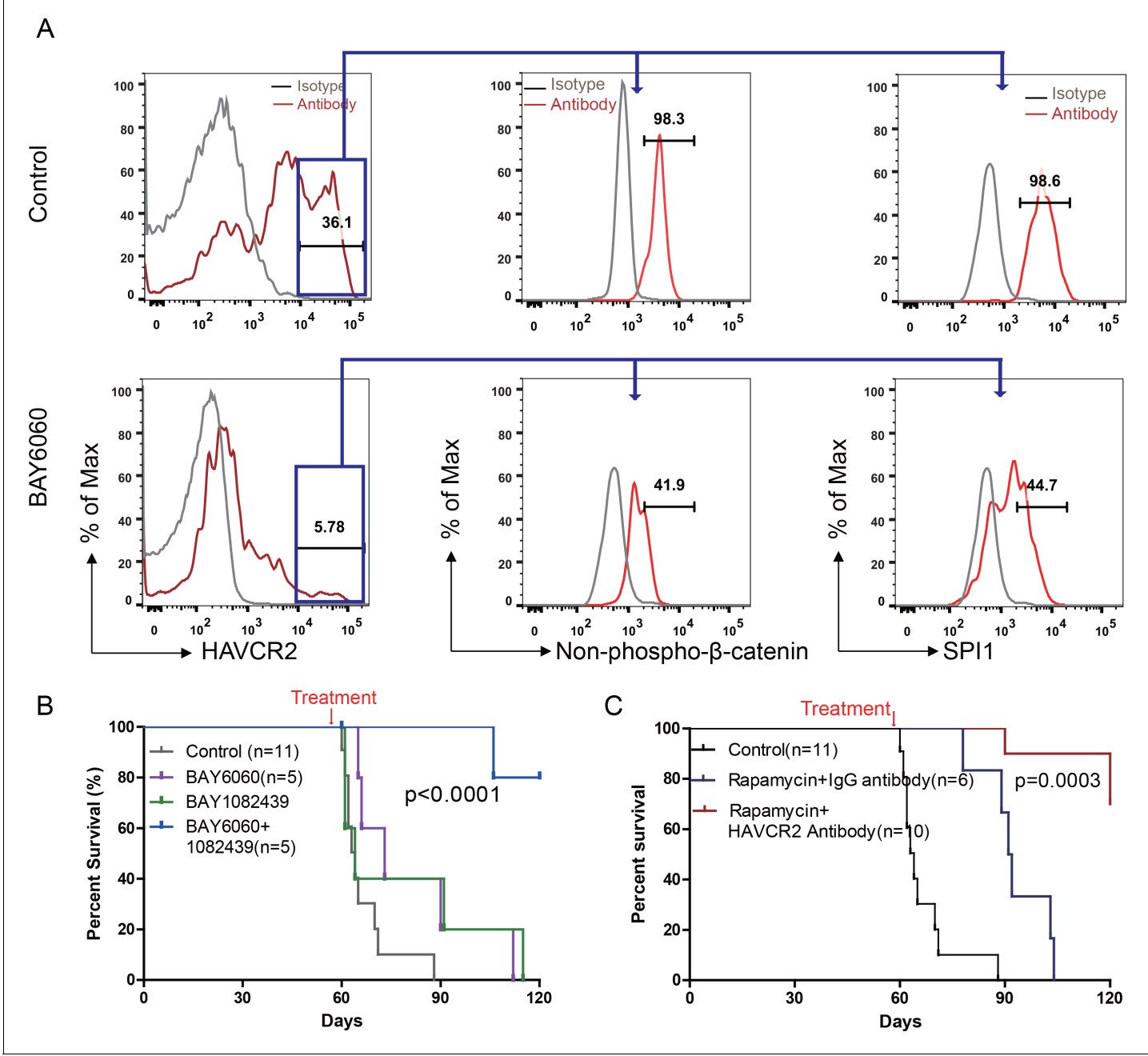

**Figure 12.** Cotargeting oncogenic driver mutations and LSC 'stemness' maintenance circuit. (A) Comparison of the HAVCR2[high] subgroup population (left panel) and of the levels of non-phosphorylated β-catenin (middle panel) and SPI1 (right panel) within the HAVCR2[high] subgroup without (upper panels) and with BAY6060 treatment (low panels); (B) Survival curve for *Cdh5-Cre*[+];*Pten*[L/L] mice treated with BAY6060 and BAY1082439 alone and in combination; (C) Survival curve for *Cdh5-Cre*[+];*Pten*[L/L] mice treated with rapamycin in combination with either an IgG control antibody or an anti-HAVCR2 antibody.

DOI: https://doi.org/10.7554/eLife.38314.018

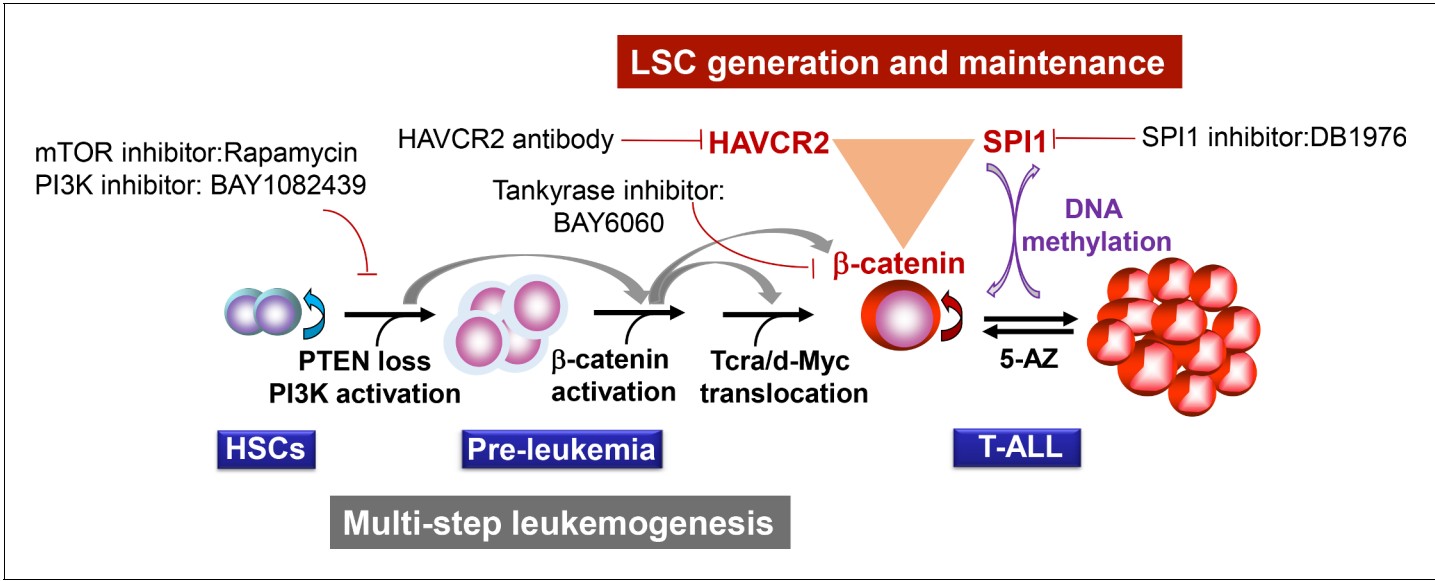

**Figure 13.** Two-layer control mechanisms for leukemogenesis and LSC maintenance.
DOI: https://doi.org/10.7554/eLife.38314.019

# Materials and methods

**Key resources table**

| Reagent type (species) or resource | Designation | Source or reference | Identifiers | Additional information |
|---|---|---|---|---|
| Strain, strain background (*Mus musculus*) | *Cdh5-Cre$^+$; Pten$^{loxP/loxP}$; Rosa26$^{floxedSTOP}$-LacZ$^+$* | (*Guo et al., 2008*) | | |
| Strain, strain background (*Mus musculus*) | *Spi1$^{loxP/loxP}$* | (*Dakic et al., 2005*) | | |
| Strain, strain background (*Mus musculus*) | *Spi1-GFP* | (*Nutt et al., 2005*) | | |
| Strain, strain background (*Mus musculus*) | *Pten/Spi1 double KO* | This Paper | | |
| Cell line (*Homo sapiens*) | KE-37 | Deutsche Sammlung von Mikroorganismen und Zellkulturen (DSMZ) | ACC-46, RRID:CVCL_1327 | |
| Cell line (*Homo sapiens*) | Jurkat | (*Schubert et al., 2014*) | | Received Drs. G. Cheng at UCLA |
| Cell line (*Homo sapiens*) | CEM | (*Schubert et al., 2014*) | | Received C. Radu at UCLA |
| Cell line (*Homo sapiens*) | HEK 293T | American Type Culture Collection (ATCC) | CRL-3216, RRID:CVCL_0063 | |
| Cell line (*Mus musculus*) | HE001 | (*Schubert et al., 2014*). | | |

*Continued on next page*

*Continued*

| Reagent type (species) or resource | Designation | Source or reference | Identifiers | Additional information |
|---|---|---|---|---|
| Antibody | TER119-APC-Cy7 | Biolegend | 116223 | |
| Antibody | B220-APC-Cy7 | Biolegend | 103224 | |
| Antibody | CD45-PE | Biolegend | 103108 | |
| Antibody | CD3-PE-Cy7 | Biolegend | 100320 | |
| Antibody | c-Kit-APC | Biolegend | 105812 | |
| Antibody | HAVCR2-PE | Biolegend | 134004 | |
| Antibody | ITGAX-FITC | Biolegend | 117306 | |
| Antibody | Mac-1-PB | Biolegend | 101224 | |
| Antibody | Gr-1-APC | Biolegend | 108412 | |
| Antibody | SPI1-PE | Biolegend | 681308 | |
| Antibody | MYC | Cell Signaling Technology | 5605S | |
| Antibody | Nonphospho (active)-β-catenin | Cell Signaling Technology | 70034 s | |
| Antibody | rabbit IgG | Cell Signaling Technology | 3900 s | |
| Antibody | Phospho-NF-κB p65 | Cell Signaling Technology | 3033 s | |
| Antibody | SPI1 | Cell Signaling Technology | 2258 s | |
| Antibody | Phospho-GSK-3β | Cell Signaling Technology | 5558 s | |
| Antibody | HAVCR2 | Abcam | ab185703 | |
| Antibody | HAVCR2 | BioxCell | RMT3-23 | |
| Antibody | IgG | BioxCell | 2A3 | |
| Antibody | Fluorescein (FITC) AffiniPure Fab Fragment Donkey Anti-Rabbit IgG (H + L) | Jackson Immuno Research | 711-097-003 | |
| Chemical compound, drug | Rapamycin | LC laboratories | R-5000 | |
| Chemical compound, drug | DB1976 | (*Belver and Ferrando, 2016*) | | |
| Chemical compound, drug | 5-AZ | Selleck | S1782 | |
| Chemical compound, drug | BAY10 82439 | (*Hill et al., 2017*) | | Provided by Bayer Pharma ceuticals |
| Chemical compound, drug | BAY 6060 | This paper | | Provided by Bayer Pharma ceuticals |

## Mice

The *Cdh5-Cre+;PtenloxP/loxP;Rosa26floxedSTOP-LacZ+* floxedSTOP-LacZ line was described previously (*Guo et al., 2008*; *Guo et al., 2011*; *Schubbert et al., 2014*). The *Spi1loxP/loxP* and *Spi1-GFP* mouse lines were kindly provided by Dr. Stephen L. Nutt. Mouse genotypes were determined by genomic PCR analyses with the primer sets listed in *Supplementary File 1*. Animal housing, breeding, and surgical procedures were approved by the Ethics Committee under ID LSC-WuH-1 and conducted in accordance with the regulations of the Division of Laboratory Animal Medicine at Peking University.

## Cell lines

The KE-37 human T-ALL cell line was purchased from DMSZ, CEM and Jurkat cell lines were generously provided by C. Radu and Drs. G. Cheng at UCLA, respectively. All of the human T-ALL cell lines were maintained in 1640 (Life Technologies) supplemented with 10% FBS, penicillin, and streptomycin. The *Pten*-null T-ALL cell line (HE001) was generated previously reported, and cultured in DMEM (Life Technologies) added with 20% FBS(Omega Scientific), 10 ng/mL IL-2, and 10 ng/mL IL-7 (both Invitrogen), 10 mmol/L HEPES, nonessential amino acids, sodium pyruvate, glutamine, penicillin, and streptomycin (Life Technologies), and 2-mercaptoethanol ($\beta$-ME; Sigma)(*Schubbert et al., 2014*). All cell lines were maintained according to the manufacturer recommendations or previous publications. CEM, Jurkat, HEK293, and KE-37 cells were authenticated by the providers and independently authenticated (via Hi-C, WES and RNAseq analyses for genome-wide alteration, mutation signatures and gene expression profiles) in the lab. All lines tested negative for mycoplasma.

## Fluorescence-activated cell sorting (FACS) analyses

FACS analyses were performed on BD LSR Fortessa or Influx system from BD Biosciences. The numbers of leukemia blasts, LSC-enriched subpopulations, and HAVCR2/ITGAX subgroups, as well as intracellular protein levels, were analyzed as described previously (*Guo et al., 2008*; *Guo et al., 2011*; *Schubbert et al., 2014*).

## Bulk RNA-seq analysis

For bulk RNA-seq analysis, total RNA was extracted from FACS-sorted cells using a RNeasy Micro Kit (Qiagen, 74004). Strand-specific libraries were generated using an NEBNext Ultra RNA Library Prep Kit (NEB, E7530) following the manufacturer's protocol. Libraries of 350±20 bp were obtained, and the quality was determined using a Fragment Analyzer system (Advanced Analytical).

Barcoded libraries were subjected to 150 bp paired-end sequencing on an Illumina HiSeq 2500, and the paired-end reads were aligned to the mouse reference genome (Version mm9 from UCSC) using Tophat (v2.0.13)(*Trapnell et al., 2009*). The expression value was generated as the number of fragments per kilobase of transcript per million mapped reads (FPKM) using Cufflinks (v2.2.1) (*Trapnell et al., 2012*).

## Single-cell RNA-seq analysis

For single-cell RNA-seq analysis, we essentially followed a published protocol (*Li et al., 2017*). Raw reads were processed as previously reported (*Li et al., 2017*; *Trapnell et al., 2009*) to generate expression values. Low-quality cells with less than 10,000 reads or less than 3000 covered genes were filtered out. Genes with a mean expression (TPM) value of less than one were discarded, leaving 276 cells and 12972 genes for further analysis. The unique gene set was then used for PCA, t-SNE, and pseudotime analyses (*Qiu et al., 2017a*; *Qiu et al., 2017b*; *Trapnell et al., 2014*). Differentially expressed genes were identified by SCDE (*Fan et al., 2016*; *Kharchenko et al., 2014*), and genes with Z > 4 were selected. Gene Ontology analysis was performed by Cluster Profiler (*Yu et al., 2012*), followed by Gene Set Enrichment Analysis (GSEA)(*Subramanian et al., 2005*) to identify gene sets that show significant differences between the blast and HAVCR2high subgroups.

## Transplantation assay

*Pten-null* T-ALL cells harvested from primary *Pten-null* T-ALL mice were FACS-sorted and diluted before transplantation, as described previously (*Guo et al., 2008*). Leukemia development was monitored daily by physical appearance, and weekly by peripheral blood smear and FACS analysis..

T-ALL was confirmed if the bone marrow or peripheral blood contained 20% leukemic blasts (*Guo et al., 2008*).

For human T-ALL cell transplantation, CEM cells were treated with 5 µM 5-AZ or PBS for 6 days *in vitro*, and an equal number of untreated and treated cells were then transplanted by tail vein injection into NSG recipients.

## Real-time PCR

Total RNA was isolated using the RNeasy Micro Kit (Qiagen, 74004) and was reverse transcribed into cDNA using a HiScript II Q RT SuperMix for qPCR Kit (Vazyme, R223-01). Gene expression levels were measured with quantitative real-time PCR using a HiScript II One Step RT-PCR Kit (Vazyme, P611-01) and a CFX Real-Time PCR detection system (Bio-Rad). All expression data were normalized to $\beta$-actin expression, and the relative expression levels were derived from the delta-delta Ct values using CFX software (Bio-Rad). For the primer sequences used, please see *Supplementary File 2*.

## Plasmid construction

The full-length *Spi1* sequence was PCR-amplified from cDNAs generated from HAVCR2[high] cells (primers: EcoRI-SPI1-Forward 5'-GAATTCATGTTACAGGCGTGCAAAATGGAAG-3' and XhoI-SPI1-Reverse 5'-CTCGAGTCAGTGGGGCGGGAGGCG-3'). The PCR products were purified and cloned into the MSCV-IRES-EGFP vector, generously provided by Dr. Owen Witt of UCLA, and the sequence was confirmed. The pll3.7-shSPI1 and control constructs were kindly provided by Dr. Junwu Zhang of the Chinese Academy of Medical Sciences and Peking Union Medical College, PLVX-IRES-RFP plasmid and PLVX-active-β-catenin (S33A, S37A, S45A) plasmid were kindly provided by Dr. Wei Guo of Tsinghua University.

## Western blot analysis

To quantify the protein levels of MYC and SPI1, Western blotting was performed as described previously (*Schubbert et al., 2014*) and the membranes were probed with antibodies against MYC (5605s),and SPI1(2258s) from Cell Signaling Technology, using HAVCR2 (ab185703) antibody from abcam, $\beta$-actin (7210,Santa Cruz) as a loading control.

## Inhibitor and antibody treatments

Two-month-old *Pten*-null T-ALL leukemic mice were treated with 1) a daily dose of rapamycin (4 mg/kg, i.p; LC laboratories), DB1976 (2.5 mg/kg, oral; synthesized by Dr. Lei's laboratory), or a combination of the two drugs; 2) a daily dose of BAY1082439 (75 mg/kg, oral; provided by Bayer Company), BAY6060(10 mg/kg, oral; provided by Bayer Company), or a combination of the two drugs; and 3) a daily dose of rapamycin (4 mg/kg, i.p; LC Laboratories) with twice weekly IgG (200 µg, i.p; 2A3,BioxCell) control or monoclonal anti-HAVCR2 (200 µg, i.p; RMT3-23,BioxCell) antibody. The durations of the treatments are indicated in *Figure 11—figure supplement 1*. HE and immunohistochemical (IHC) analyses were performed as described (*Guo et al., 2008*).

For 5-AZ (S1782, Selleck) treatment, 6-week-old *Pten*-null T-ALL mice were treated with either vehicle or 5-AZ (1.25 mg/kg, i.p.; 3 days per week) for 2 weeks before intracellular FACS analysis.

## RRBS library preparation

Blast- and LSC-enriched subpopulations were collected by FACS sorting, and genomic DNA was extracted using a DNA micro kit or a DNA mini kit (Qiagen). The RRBS library was prepared according to a previous publication (*Smallwood and Kelsey, 2012*). Genomic DNA was digested with MspI (Fermentas), followed by end repair, adapter ligation and bisulfite modification (Qiagen, #59104). The converted DNA library was sequenced on a HiSeq 4000 (Illumina) after two-round PCR amplification and size selection.

## DNA methylation analysis

BS-seq reads were aligned to the reference genome (mm9) by BS-Seeker2 (*Guo et al., 2013*). The lollipop plot and region-specific distribution profiles were generated by CGmap Tools (*Guo et al., 2018*). The methylation status of murine and human SPI1 promoter CpG islands was determined according to (*Fernández-Nestosa et al., 2013*).

## Conventional bisulfite sequencing

For bisulfite conversion, genomic DNA was treated with an EZ-DNA Methylation-Direct Kit (D5021, Zymo Research) according to the manufacturer's protocol. The converted DNA was subjected to PCR amplification and cloned into a pEASY-T1 Simple cloning vector (Transgene Biotech). The bisulfite primers for the mouse and human promoters were described previously (*Fernández-Nestosa et al., 2013*; *Tatetsu et al., 2007*)(Table S2). Individual clones were sequenced by Sanger sequencing, and the data were analyzed by the online software Quma (http://quma.cdb.riken.jp).

## Chromatin immunoprecipitation (ChIP)

Approximately $5 \times 10^6$ Blast-EGFP and Blast-SPI1 cells were used, and ChIP analysis was performed using a Zymo-Spin ChIP Kit (D5210, Zymo Research). The antibodies used for the ChIP assays were anti-SPI1 (sc-352, Santa Cruz) and normal rabbit IgG (2729, Cell Signaling Technology). The enriched regions were quantified by qPCR using the primers described in *Supplementary File 2*.

## Data access

All the Bulk RNA-seq, Single cell RNA-seq and BiSulfite-seq data for this study are deposited in NCBI Gene Expression Omnibus (GEO; accession # GSE115356; https://www.ncbi.nlm.nih.gov/geo/query/acc.cgi?acc=GSE115356).

## Statistical analysis

GraphPad Prism software was used to calculate the means and standard deviations (SDs). The *t*-test or two-way ANOVA was used to determine statistical significance, and $p < 0.05$ was considered statistically significant. The data are presented as the means ± SDs.

## Additional information

### Competing interests

Ningshu Liu: is an employee of Bayer AG. The other authors declare that no competing interests exist.

### Funding

| Funder | Grant reference number | Author |
| --- | --- | --- |
| National Natural Science Foundation of China | 81602254 | Lu Yang |
| National Key Research | 2017YFA0505200 | Xiaoguang Lei |
| National Natural Science Foundation of China | 21472010 | Xiaoguang Lei |
| National Natural Science Foundation of China | 21561142002 | Xiaoguang Lei |
| National Natural Science Foundation of China | 21625201 | Xiaoguang Lei |
| National Natural Science Foundation of China | 81570118 | Xiaoguang Lei |
| Center for Life Sciences | | Hong Wu |
| Beijing Advanced Innovation Center for Genomics | | Hong Wu |
| Bayer | | Hong Wu |

The funders had no role in study design, data collection and interpretation, or the decision to submit the work for publication.

## Author contributions

Haichuan Zhu, Conceptualization, Data curation, Formal analysis, Investigation, Visualization, Methodology, Writing—original draft, Writing—review and editing; Liuzhen Zhang, Yilin Wu, Weilong Guo, Data curation, Formal analysis, Investigation, Writing—review and editing; Bingjie Dong, Mei Wang, Software, Formal analysis, Writing—review and editing; Lu Yang, Investigation, Writing—review and editing; Xiaoying Fan, Yuliang Tang, Investigation; Ningshu Liu, Investigation, Writing—review and editing, Providing BAY1082439 and BAY6060; Xiaoguang Lei, Supervision, Funding acquisition, Project administration, Writing—review and editing; Hong Wu, Conceptualization, Resources, Supervision, Funding acquisition, Writing—original draft, Project administration, Writing—review and editing

## Author ORCIDs

Weilong Guo ⓘ http://orcid.org/0000-0001-5199-1359
Mei Wang ⓘ http://orcid.org/0000-0003-3292-1413
Hong Wu ⓘ http://orcid.org/0000-0001-7545-7919

## Ethics

Animal experimentation: All experimental protocols were approved by the Peking University Animal Care and Use Committee (IACUC).This study were approved by the Peking University Animal Care and Use Committee (LSC-WuH-1).

## Decision letter and Author response

Decision letter https://doi.org/10.7554/eLife.38314.032
Author response https://doi.org/10.7554/eLife.38314.033

# Additional files

## Supplementary files

• Supplementary File 1. Primers used for genotyping in this study.
DOI: https://doi.org/10.7554/eLife.38314.020

• Supplementary File 2. Primers used for q-PCR in this study.
DOI: https://doi.org/10.7554/eLife.38314.021

• Supplementary File 3. List of FPKM for Bulk RNA-seq in T-ALL and genes list of the yellow module reported in the *Figure 1A*.
DOI: https://doi.org/10.7554/eLife.38314.022

• Supplementary File 4. The genes list of the HAVCR2$^{high}$ signature genes (1079) and SPI1 target genes (710) reported in the *Figure 4A*.
DOI: https://doi.org/10.7554/eLife.38314.023

• Transparent reporting form
DOI: https://doi.org/10.7554/eLife.38314.024

## Data availability

All the Bulk RNA-seq, Single cell RNA-seq and BiSulfite-seq data for this study are deposited in NCBI Gene Expression Omnibus under the accession number GSE115356.

The following dataset was generated:

| Author(s) | Year | Dataset title | Dataset URL | Database and Identifier |
|---|---|---|---|---|
| Dong B | 2018 | T-ALL Leukemia Stem Cell | https://www.ncbi.nlm.nih.gov/geo/query/acc.cgi?acc=GSE115356 | NCBI Gene Expression Omnibus, GSE115356 |

The following previously published datasets were used:

| | | | | Database and |
|---|---|---|---|---|

| Author(s) | Year | Dataset title | Dataset URL | Identifier |
|---|---|---|---|---|
| Van Vlierberghe P, Ambesi-Impiombato A, Perez-Garcia A, Haydu JE, Rigo I, Hadler M, Tosello V, Della Gatta G, Paietta E, Racevskis J, Wiernik PH, Luger SM, Rowe JM, Rue M, Ferrando AA | 2011 | Gene Expression Profile of 57 human T-ALL samples collected in human clinical trial E2993 | https://www.ncbi.nlm.nih.gov/geo/query/acc.cgi?acc=GSE33469 | NCBI Gene Expression Omnibus, GSE33469 |

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
