## [Decision Letter]

Thank you for submitting your article "T-ALL Leukemia Stem Cell "stemness" is Epigenetically Controlled by the Master Regulator PU.1" for consideration by *eLife*. Your article has been reviewed by three peer reviewers, one of whom is a member of our Board of Reviewing Editors, and the evaluation has been overseen by Sean Morrison as the Senior Editor and Thomas Look as Reviewing Editor. The reviewers have opted to remain anonymous.

The reviewers have discussed the reviews with one another and the Reviewing Editor has drafted this decision to help you prepare a revised submission.

Summary:

The manuscript by Wu and coworkers explores the mechanisms driving leukemia initiating cell activity in a mouse model of T cell lymphoblastic leukemia driven by loss of the *Pten* tumor suppressor gene in hematopoietic progenitors. This model characteristically acquires *Myc* activating chromosomal rearrangements during disease progression and leukemia growth becomes uncoupled with PI3K signaling. Early work had identified enrichment of leukemia initiating cell activity in Lin^-^CD3^+^Kit^mid^. Now transcriptomic analyses of this population compared to blasts identifies a transcriptional program enriched in surface markers and with prominent expression of TIM-3 and CD11c. Using these markers the authors can further enrich for the LIC compartment and refine the LIC-associated transcriptional signature. PU.1 emerges from these analyses as a potential master regulator of this transcriptional program and LIC activity directly linked with the regulation of TIM-3. A mechanistic link is proposed between PI3K signaling, β-catenin activation, PU.1 upregulation, TIM-3 expression as initiating cascade in leukemia development. Later on, a PI3K independent loop involving β-catenin, *Pu.1* and *TIM-3* maintains this program in LICs. PU.1 methylation in blast cells is shown as a potential mechanism for abrogating the activity of this program in blast cells. Therapeutically, inhibition of PU.1 with DB1976, an anti-TIM-3 antibody and rapamycin or BAY1082439 for debulking blast cells induces profound antileukemic effects.

This is an interesting, mature and clearly presented manuscript that dissects in great depth the mechanisms of LIC activity operating in mouse tumors generated by hematopoietic specific inactivation of *Pten*. Key strengths of the work as presented include the compelling phenotypes resulting from genetic inactivation of *Pu.1* in the *Pten* knockout mouse model and the profound antileukemic effects observed with inhibitors targeting the regulatory circuit implicated here in the maintenance of LIC activity.

While overall the findings are intriguing, the reviewers have suggestions to correct gaps in the experimental evidence provided. If the authors can address the issues noted below this could be a very interesting article.

1) Enthusiasm for the manuscript is limited by the lack of a compelling case for the relevance of the mechanisms proposed in human disease. KE-37 and CEM cell lines are analyzed as examples of PU.1^+^ and PU.1^-^ leukemia to illustrate the regulatory role of PU.1 on TIM-3 and CD11c, yet it is unclear that PU.1 expression or knockdown modifies the LIC activity in these lines or whether these lines really still have true LIC. Attention should be paid to the presence of MYC translocations and PTEN mutations in these lines. Correlative data on PU.1 TIM-3 and LMO2 in primary T-ALL samples is of interest but as the authors correctly and insightfully point out this correlation may reflect primarily developmental programs, which are potentially but not necessarily linked with LIC activity.

Evaluation of the LIC activity in primary human T-ALL xenografts (PDX models) following the TIM-3 CD11c markers and ideally testing the effects of PU.1 inhibition in human T-ALL LIC activity would greatly enhance the relevance and general interest of the results presented here. We ask this in the hope that the authors may already have data in human T-ALL models, because this type of data would take more than a few months to generate, which is a goal of *eLife*. If such data is not available, the authors should at least point out the need for such studies and acknowledge the limitations of established T-ALL cell lines for studying the biology of LSC populations.

2) In Figure 3D, the authors use PU.1-EGFP for PU.1 ChIP analysis. Was it not possible to do the ChIP-PU.1 experiment without introducing PU.1-EGFP? Isn't there an antibody suitable to detect PU.1 in ChIP-seq? The authors should do a straightforward ChIP-seq experiment for endogenous PU.1 and detect the genes that bind PU.1 in the TIM-3^high^ mouse leukemia cell population. ChIP-PCR is subject to bias and it would be nice to show which of the 710 possible PU.1 target genes are actually bound by PU.1?

3) Figure 3H. Please include Western blot for TIM-3.

4) Figure 5C. β-catenin is also known to upregulate MYC and yet the TIM-3^high^PU.1^high^ LSC T-ALL subpopulation appears to have low levels of MYC expression due to PU.1 activation. Could the authors include *Myc* expression levels in Figure 5C? Does the *Myc* expression level go up or down when β-catenin is becomes activated by TIM-3?

5) The authors propose a feed forward loop between PU.1/TIM-3 that is further regulated by active β-catenin, but the key data presented (Figure 5) are correlative and do not formally demonstrate that this is either a feed forward loop (i.e. that TIM-1 regulates PU.1 as well as the converse; nor that these are directly regulated by β-catenin). This needs to be strengthened or the concept of a feed forward loop should be downplayed in the interpretation.

Stylistic comments:

1) *eLife* does not limit the number of figures. Also each figure can have linked supplemental figures. In the current paper, each of the figures has too many panels, such that the panels are very small and the font size is much too small. These panels should be broken up, with an increased number of main figures and figure supplements so that the results are more accessible. Please label the graphs with bigger font size – it is hard to read the labels.

2) Figure 2E is a table in a figure, and lists a lot of sensitivity, resistance data etc. that are not described.

3) Overall the English in this paper is very understandable, but grammatical mistakes are found throughout the text. A scientific editor familiar with English grammar could very quickly improve the English grammar and clarity of the paper.

---

## [Author Response]

Please note some of the gene names, such as PU.1, TIM-3, CD11c, have been changed to approved gene names systematically per editorial suggestions.

[…] While overall the findings are intriguing, the reviewers have suggestions to correct gaps in the experimental evidence provided. If the authors can address the issues noted below this could be a very interesting article.1) Enthusiasm for the manuscript is limited by the lack of a compelling case for the relevance of the mechanisms proposed in human disease. KE-37 and CEM cell lines are analyzed as examples of PU.1^+^ and PU.1^-^ leukemia to illustrate the regulatory role of PU.1 on TIM-3 and CD11c, yet it is unclear that PU.1 expression or knockdown modifies the LIC activity in these lines or whether these lines really still have true LIC. Attention should be paid to the presence of MYC translocations and PTEN mutations in these lines. Correlative data on PU.1 TIM-3 and LMO2 in primary T-ALL samples is of interest but as the authors correctly and insightfully point out this correlation may reflect primarily developmental programs, which are potentially but not necessarily linked with LIC activity.

We appreciate reviewers’ critical comments on the human T-ALL cell lines used in our study. As indicated in the table below, KE-37 is *PTEN* null, *NOTCH1*WT and carries *TRAD-MYC* translocation, very similar to the genetic alterations associated with the *Pten* null T-ALL model used in our study, while CEM is *PTEN* null, *NOTCH* mutated and WT for *c-MYC* (Milani et al., 2018; Schubbert et al., 2014). We used these two lines to validate (1) the transcription regulation of the SPI1 target genes in human T-ALL (revised Figure 4I); and (2) the epigenetic regulation of the human *SPI1* expression (revised Figure 10). Our recent analysis also indicates that the expression levels of SPI1 and HAVCR2 are higher in human ETP T-ALL as compared to non-ETP T-ALL (Author response image 1). We concur that cell lines are most suitable for studying signaling and regulatory mechanism but not LIC activity.

Author response table 1.

PTENNOTCH1Myc*Pten* null T-ALLNullWT*Tcra/b-Myc*KE-37NullWT*TRAD-MYC*CEMNullMutWT

Evaluation of the LIC activity in primary human T-ALL xenografts (PDX models) following the TIM-3 CD11c markers and ideally testing the effects of PU.1 inhibition in human T-ALL LIC activity would greatly enhance the relevance and general interest of the results presented here. We ask this in the hope that the authors may already have data in human T-ALL models, because this type of data would take more than a few months to generate, which is a goal of eLife. If such data is not available, the authors should at least point out the need for such studies and acknowledge the limitations of established T-ALL cell lines for studying the biology of LSC populations.

We totally agree with reviewer’s comment and have indeed generated six T-ALL PDX models. Unfortunately, none of PDXs growing up in the NSG mice so far belong to ETP T-ALL subtype, which our model closely mimics to (revised Figures 7 and 8). We have now collected more bone marrow samples and transplanted equal number of ETP and non-ETP T-ALL cells into NSG mice. Our preliminary study reveals that non-ETP cells may have better reconstitution ability than ETP T-ALL cells in NSG mice (Author response image 1). Since ETP T-ALL is originated from early marrow or thymic progenitor cells before T cell commitment and its development depends on the functional thymic environment, NSG mice may not be suitable for studying human ETP T-ALL LICs without certain supplements or pre-requisitions, such as PTEN loss, which enables T cell differentiation in the absence of the recombination-activating gene (Reg) and substitutes for both IL-7 and pre-TCR signals (Guo et al., 2011; Hagenbeek et al., 2004).

**Author response image 1. respfig1:** ETP and non-ETP T-ALL have different HAVCR2 and SPI1 expression levels and leukemia reconstitution abilities. (**A**). The HAVCR2 and SPI1 expression levels are significant higher in ETP T-ALL samples as compared to non-ETP T-ALL samples. ***, p<0.001. (**B**). Bone marrow cells (1X10^6^) from human T-ALL patients were *i.v.* injected into the NSG mice. Leukemia reconstitution was determined when >25% of the cells in the peripheral blood were human CD45+;CD7+ leukemic cells.

There are very few publications on T-ALL LIC and T-ALL PDX models. The majority of these published works are based on non-ETP subtype and their conclusions about LIC are not consistent with each other (Chiu et al., 2010; Cox et al., 2007). We are conducting more detailed LIC-enrichment studies as well as genomic analyses of all T-ALL samples we have in hand. In the meantime, we are also exploring *in vitro* culture systems.

Since these works will take more than several months to accomplish, we have revised our manuscript and acknowledged the limitations of established human T-ALL cell lines for studying the biology of LICs (Subsection “LSCs loses their “stemness” when *Spi1* expression is silenced by DNA methylation”, second paragraph and Discussion, last paragraph).

2) In Figure 3D, the authors use PU.1-EGFP for PU.1 ChIP analysis. Was it not possible to do the ChIP-PU.1 experiment without introducing PU.1-EGFP? Isn't there an antibody suitable to detect PU.1 in ChIP-seq? The authors should do a straightforward ChIP-seq experiment for endogenous PU.1 and detect the genes that bind PU.1 in the TIM-3^high^ mouse leukemia cell population. ChIP-PCR is subject to bias and it would be nice to show which of the 710 possible PU.1 target genes are actually bound by PU.1?

The frequency of HAVCR2^high^ subpopulation is about 1/10000 while regular ChIP-seq requires 10^6-7^ cells. Since we can only sort about 4000 cells/hour, it is technically difficult to do the ChIP-seq analysis on endogenously isolated HAVCR2^high^ subpopulation. Our attempts of expanding HAVCR2^high^ subpopulation *in vitro* without changing its LSC activity have not been successful.

We also like to clarify the path that leads to the discovery of SPI1 in our study, which was not clearly described in our initial submission. We first used network component analysis, in which the activity of transcription factors can be deduced based on the expression levels of their target genes. Among the predicted transcription factors that may control the expression of HAVCR2^high^ LSC signature genes, SPI1 scores the highest. We then compared 1079 HAVCR2^high^ LSC-signature genes to 7180 T cell development related SPI1 target genes determined by the endogenous SPI1 ChIP-seq analysis (Zhang et al., 2012) and found 710 overlapping genes (please see revised description in the first paragraph of the subsection “SPI1 is the master regulator of LSC signature genes”).

3) Figure 3H. Please include Western blot for TIM-3.

We have now included Western blot for HAVCR2 in our revised Figure 4H.

4) Figure 5C. β-catenin is also known to upregulate MYC and yet the TIM-3^high^PU.1^high^ LSC T-ALL subpopulation appears to have low levels of MYC expression due to PU.1 activation. Could the authors include Myc expression levels in Figure 5C? Does the Myc expression level go up or down when β-catenin is becomes activated by TIM-3?

As shown in the revised Figure 8A (original Figure 5C), *Myc* expression is indeed downregulated upon β-catenin activation in this *in vitro* system. MYC protein levels are also negatively correlated with the levels of non-phosphorylated β-catenin and HAVCR2 *in vivo*(revised Figure 2B and Figure 8C).

5) The authors propose a feed forward loop between PU.1/TIM-3 that is further regulated by active β-catenin, but the key data presented (Figure 5) are correlative and do not formally demonstrate that this is either a feed forward loop (i.e. that TIM-1 regulates PU.1 as well as the converse; nor that these are directly regulated by β-catenin). This needs to be strengthened or the concept of a feed forward loop should be downplayed in the interpretation.

We now have additional data to support the role of HAVCR2 in SPI1 regulation:

1) Treating the *Pten* null T-ALL model with the anti-HAVCR2 antibody for 2 days is sufficient to reduce HAVCR2^high^ subpopulation, accompanied by reduced non-phosphorylated β-catenin levels *in vivo*, similar to the effects of tankyrase/β-catenin inhibitor BAY6060 shown in revised Figure 8G and Figure 12A.

2) Gal-9 is a ligand of HAVCR2. Previous studies indicate that Gal-9 is critical for HAVCR2 signaling and human AML LSC self-renewal (Kikushige et al., 2015). We have genetically deleted *Gal-9* and found that Gal-9 loss prevents the *Pten* null T-ALL development and HAVCR2^high^ LSC formation in vivo, an effect similar to that of *Spi1* deletion (Author response image 2).

**Author response image 2. respfig2:** Gal-9 is essential for the *Pten* null T-ALL LSC formation and leukemia development. (**A**). Deletion of the HAVCR2 ligand Gal-9 delays or prevents the *Pten* null T-ALL development *in vivo*. (**B**). HAVCR2^high^ LSCs at ETP stage in the *Pten* null T-ALL are eliminated by *SPI1* or *Gal-9* deletion *in vivo*.

Nevertheless, we do agree with reviewer’s critical comment since β-catenin-SPI1-HAVCR2 forms an inter-dependent regulatory circuit and lacking anyone of them will influence LSC generation and self-renewal, as indicated by our *Pten;Spi1* and *Pten;Gal-9* knockouts and combination treatment experiments. In our revised Figure 13, we replaced feed forward arrows with a shaded triangle to more accurately present our take-home message. We also replaced “feedback loop” with “regulatory circuit” in our revised manuscript.

Stylistic comments:

*1)* eLife *does not limit the number of figures. Also each figure can have linked supplemental figures. In the current paper, each of the figures has too many panels, such that the panels are very small and the font size is much too small. These panels should be broken up, with an increased number of main figures and figure supplements so that the results are more accessible. Please label the graphs with bigger font size – it is hard to read the labels.*

We have revised figures per your instruction and the revised manuscript now contains 13 figures, 1 table and 4 supplementary figures:

– Revised Figure 1 is the original Figure 1.

– Revised Figures 2-3 and Table 1 are from the original Figure 2.

– Revised Figures 4 – 5 are from the original Figure 3.

– Revised Figure 6 is the original Figure 4.

– Revised Figures 7-8 are from the original Figure 5 and supplementary Figure 4.

– Revised Figures 9–10 are from the original Figure 6 and supplementary Figure 5.

– Revised Figures 11–13 are from the original Figure 7 and supplementary Figure 6.

2) Figure 2E is a table in a figure, and lists a lot of sensitivity, resistance data etc. that are not described.

The original Figure 2E has been moved to Table 1 with a clear description.

3) Overall the English in this paper is very understandable, but grammatical mistakes are found throughout the text. A scientific editor familiar with English grammar could very quickly improve the English grammar and clarity of the paper.

Thanks for pointing out this deficiency. The revised manuscript has been edited by Nature Research Editing Service.

References:

Hagenbeek, T.J., Naspetti, M., Malergue, F., Garçon, F., Nunès, J.A., Cleutjens, K.B.J.M., Trapman, J., Krimpenfort, P., Spits, H. (2004) The Loss of PTEN Allows TCR αβ Lineage Thymocytes to Bypass IL-7 and Pre-TCR–mediated Signaling. J Exp Med. Oct 4; 200(7): 883–894, doi: 10.1084/jem.20040495